# The influence of vegetation water dynamics on the ASCAT backscatter-incidence angle relationship in the Amazon

Ashwini Petchiappan[1], Susan C. Steele-Dunne[2], Mariette Vreugdenhil[3], Sebastian Hahn[3], Wolfgang Wagner[3], and Rafael Oliveira[4]

[1]Department of Water Management, Delft University of Technology, Stevinweg 1, Delft 2600 GA, The Netherlands
[2]Department of Geoscience and Remote Sensing, Delft University of Technology, Stevinweg 1, Delft 2600 GA, The Netherlands
[3]Department of Geodesy and Geo-Information, TU Wien, Vienna 1040, Austria
[4]Department of Plant Biology, Institute of Biology P.O.Box: 6109, University of Campinas – UNICAMP 13083-970, Campinas, SP, Brazil

**Correspondence:** Susan Steele-Dunne (s.c.steele-dunne@tudelft.nl)

**Abstract.** Microwave observations are sensitive to plant water content and could therefore provide essential information on biomass and plant water status in ecological and agricultural applications. The combined data record of the C-band scatterometers on ERS 1/2, the Metop series and the planned Metop Second Generation satellites will span over 40 years, which would provide a long-term perspective on the role of vegetation in the climate system. Recent research has indicated that the unique viewing geometry of ASCAT could be exploited to observe vegetation water dynamics. The incidence angle dependence of backscatter can be described with a second order polynomial, the slope and curvature of which are related to vegetation. In a study limited to grasslands, seasonal cycles, spatial patterns and interannual variability in the slope and curvature were found to vary among grassland types and were attributed to differences in moisture availability, growing season length and phenological changes. To exploit ASCAT slope and curvature for global vegetation monitoring, their dynamics over a wider range of vegetation types needs to be quantified and explained in terms of vegetation water dynamics. Here, we compare ASCAT data with meteorological data and GRACE Equivalent Water Thickness (EWT) to explain the dynamics of ASCAT backscatter, slope and curvature in terms of moisture availability and demand. We consider differences in the seasonal cycle, diurnal differences, and the response to the 2010 and 2015 droughts across ecoregions in the Amazon basin and surroundings. Results show that spatial and temporal patterns in backscatter reflect moisture availability indicated by GRACE EWT. Slope and curvature dynamics vary considerably among the ecoregions. The evergreen forests, often used as a calibration target, exhibit very stable behaviour even under drought conditions. The limited seasonal variation follows changes in the radiation cycle, and may indicate phenological changes such as litterfall. In contrast, the diversity of land cover types within the Cerrado region results in considerable heterogeneity in terms of the seasonal cycle and the influence of drought on both slope and curvature. Seasonal flooding in forest and savanna areas also produced a distinctive signature in terms of the backscatter as a function of incidence angle. This improved understanding of the incidence angle behaviour of backscatter increases our ability to interpret and make optimal use of the ASCAT data record and Vegetation Optical Depth products for vegetation monitoring.

## 1 Introduction

Microwave remote sensing observations are sensitive to plant water content, which depends on above ground biomass and plant water status (Konings et al., 2019; Owe et al., 2001; Jackson et al., 1982). Data from active and passive microwave sensors can provide valuable information about vegetation in a range of applications in ecological and agricultural monitoring (Konings et al., 2019; Chaparro et al., 2019; Rao et al., 2019; Steele-Dunne et al., 2017; Tian et al., 2016; Andela et al., 2013; Saatchi et al., 2013; Liu et al., 2013; McNairn et al., 2000; Wagner et al., 1999). In particular, Vegetation Optical Depth (VOD) products derived from various passive and active microwave sensors are increasingly used for biomass monitoring (Liu et al., 2015), drought monitoring (Liu et al., 2018), wildfire risk assessment (Forkel et al., 2019) and have been related to Gross Primary Production (Teubner et al., 2018, 2019), carbon stocks (Chaparro et al., 2019) and drought-driven tree mortality (Rao et al., 2019). Currently VOD datasets are available from single sensor passive microwave observations, such as SMAP (Konings et al., 2016), SMOS (Fernandez-Moran et al., 2017) and AMSR2 (Owe et al., 2001; De Jeu, 2003), and active microwave observations such as ASCAT (Vreugdenhil et al., 2016). Furthermore, long-term data records are available that combine VOD from different sensors (Moesinger et al., 2020; Liu et al., 2011).

The current study is motivated by the availability of consistent C-band data from 1991 to at least 2030, and its potential value as a long-term data record for vegetation monitoring. The Advanced Scatterometer (ASCAT) is a real aperture radar operating at 5.255 GHz with VV polarization. There are currently three ASCAT instruments in orbit on Metop-A, Metop-B and Metop-C, launched in October 2006, September 2012 and November 2018 respectively. ASCAT builds on the success of the European Scatterometer (ESCAT) which flew on the ERS-1/2 satellites from 1991-2011 (Attema, 1991; Figa-Saldaña et al., 2002; Wagner et al., 2013)). Continuation of the ESCAT/ASCAT record is ensured by the plans to launch SCA on Metop-SG in 2024 (Stoffelen et al., 2017). Using data from a single series of satellites with identical and inter-calibrated instruments circumvents many of the challenges of reconciling data using different frequencies, viewing geometries and orbit characteristics. The continuity from ERS to Metop and Metop-SG ensures an internally consistent data product for at least 40 years, rendering it ideal to study the role of vegetation in the climate system.

Many early studies demonstrated the sensitivity of ESCAT and ASCAT backscatter to vegetation, and explored the potential value of these data for vegetation monitoring (Wismann et al., 1995; Frison et al., 1998; Woodhouse et al., 1999; Jarlan et al., 2002; Steele-Dunne et al., 2012; Schroeder et al., 2016). These studies focused on spatial and temporal variations in backscatter normalized to some reference angle. Here, the focus is on the potential information content of the incidence angle behaviour of backscatter, and particularly the so-called "Dynamic Vegetation Parameters" describing the incidence angle behaviour of backscatter as calculated in the TU Wien Soil Moisture Retrieval (TUW SMR) algorithm (Hahn et al., 2017).

The ASCAT Dynamic Vegetation Parameters refer to the parameters of the second order Taylor polynomial used to describe the incidence angle ($\theta$) dependence of backscatter $\sigma^\circ$. This is described as follows:

$$\sigma^\circ(\theta) = \sigma^\circ(\theta_r) + \sigma'(\theta) \cdot (\theta - \theta_r) + \frac{1}{2} \cdot \sigma''(\theta r) \cdot (\theta - \theta_r)^2 \qquad [dB] \tag{1}$$

where $\sigma^{\circ}(\theta_r)$, $\sigma'(\theta_r)$ and $\sigma''(\theta_r)$ are the normalized backscatter, slope and curvature at some reference angle $\theta_r$. In the TUW SMR algorithm, this expression is used to normalize backscatter values from different incidence angles to a reference angle $\theta_r$. It is also used to account for the influence of vegetation on backscatter as the incidence angle behaviour of $\sigma^{\circ}$ depends on whether total backscatter is dominated by surface scattering from the soil, volume scattering from the vegetation, or multiple scattering (Wagner et al., 1999; Naeimi et al., 2009; Hahn et al., 2017). In other words, slope and curvature are calculated and used to account for the influence of vegetation in the soil moisture retrieval. An increase in soil moisture results in an increase in backscatter at all incidence angles, while a change in the vegetation (due to growth cycle or water status) changes the sensitivity of backscatter to incidence angle, i.e. it results in a change in slope and curvature. So, the slope and curvature provide complementary information to the normalized backscatter.

Results from Steele-Dunne et al. (2019) suggest that considering the slope ($\sigma'(\theta)$) and curvature ($\sigma''(\theta)$) dynamics in combination with the backscatter could yield valuable insights into vegetation water dynamics. Seasonal cycles, spatial patterns and interannual variability in the slope varied between grassland cover type reflecting variations in soil moisture availability and growing season length. Slope is considered an indication of vegetation density, or above ground fresh biomass, which is a combination of dry biomass and vegetation water content. Results also suggested that curvature variations were influenced by the total water content, but also its vertical distribution within the vegetation and the geometry of constituents. Contiguous anomalies were observed in both slope and curvature during drought periods, suggesting that the slope and curvature provide insight into when the severity of a soil moisture anomaly is enough to impact vegetation. Diurnal variations were also observed and attributed to sub-daily variations in the dominant scattering mechanism due to changes in the vertical moisture distribution of the grasses. More recently, Pfeil et al. (2020) observed a "spring peak" in slope values around April in broadleaf deciduous forest in Europe. Using LAI and data from the Pan European Phenological database (PEP725) (Templ et al., 2018) they argued that this spring peak in ASCAT slope coincides with spring activation, particularly the increase in water content of bare twigs and branches prior to leaf out in broadleaf deciduous forests. ASCAT slope and curvature therefore seem to be sensitive to changes in vegetation water content and structure of vegetation.

The goal of this study is to improve our understanding of the ASCAT backscatter-incidence angle relationship and how they might be used to monitor vegetation water dynamics. The Amazon basin and its surroundings has been chosen as a study area as it provides a wide range in terms of expected variability in ASCAT backscatter, slope and curvature. Backscatter in the evergreen forest was considered so stable that this region has been used for satellite radar calibration (Birrer et al., 1982). In contrast, seasonal changes in the Cerrado are expected to yield strong annual cycles in backscatter, slope and curvature. Seasonal cycles and diurnal differences in ASCAT backscatter, slope and curvature will be determined for several ecoregions of interest. These will be compared to meteorological data and GRACE terrestrial water storage variations to relate the ASCAT backscatter, slope and curvature to moisture availability and demand. Finally, we will investigate whether there are anomalies in the ASCAT backscatter, slope and curvature as a result of the 2010 and 2015 droughts.

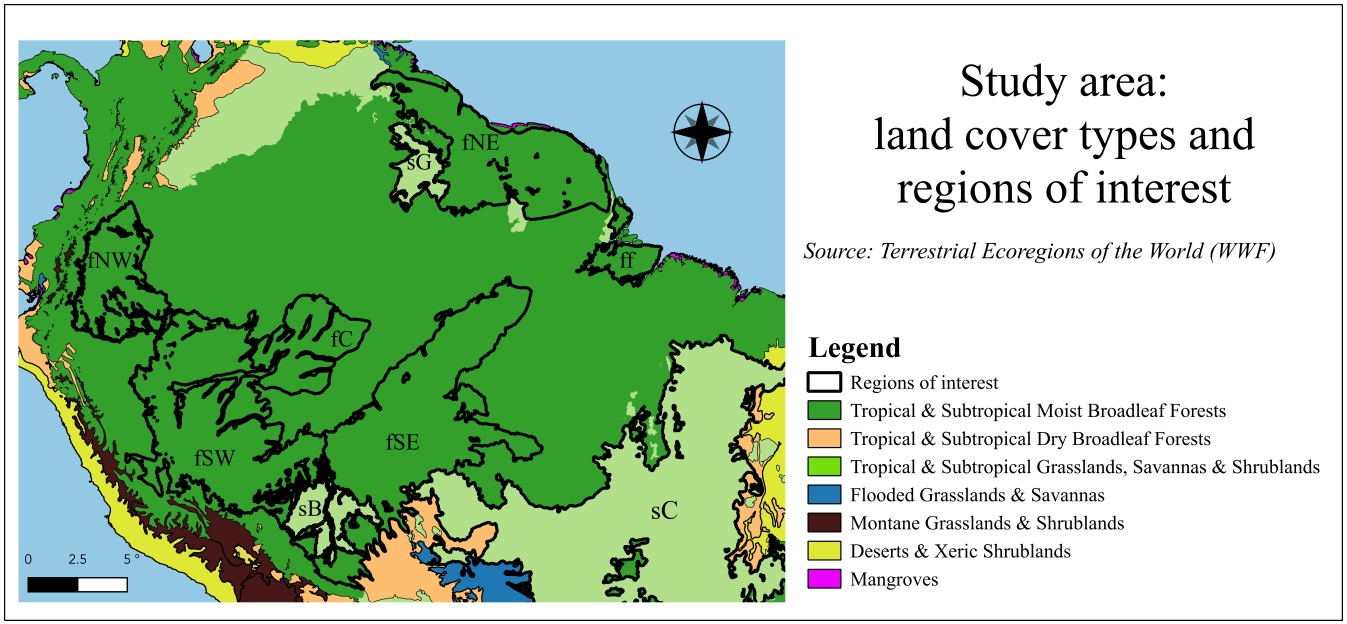

**Figure 1.** Study Area. The map is colored by biome, and nine ecoregions of interest are highlighted based on the dataset of Olson et al. (2001). The six forest ecoregions of interest are Napo moist forest (fNW), Guianan moist forests (fNE), Southwest Amazon moist forests (fSW), Madeira-Tapajos moist forest (fSE), Jurua-Perez moist forests (fC) and the Marajo Varzea flooded forests (ff). The three savanna ecoregions of interest are the Cerrado (sC), Guianan Savanna (sG) and Beni Savanna (sB).

## 2 Data and Methods

### 2.1 Study Area

Figure 1 shows the extent of the study domain, highlights the biomes (by color) and outlines the ecoregions of interest identified in the WWF Terrestrial Ecoregions dataset (WWF, 2019) and described by Olson et al. (2001). The study domain extends from 9°N to 19°S, and 44°W to 80°W. Most of the study region is covered by the Amazon rain forest, which extends over 5.3 million $km^2$ (Soares et al., 2006). Six forest ecoregions are investigated here:

1. The Napo moist forests (fNW), located in northwest Amazonia, receive some of the highest amounts of annual precipitation in the biome, reaching up to 4000 mm in some parts. This highly biodiverse region has canopies reaching 40 m.

2. The Guianan moist forests (fNE) are one of the largest continuous stretches of relatively pristine lowland tropical rainforest in the world. There are two distinct wet seasons: from December to January and from May to August. The floral diversity is rich, with multi-tiered vegetation of 40 m tall trees with herbaceous plants below. The dry season (September-November) can see a substantial reduction in leaves, although the forest is evergreen.

3. The Southwest Amazon moist forests (fSW) have significant variations in topography and soil characteristics, leading to extremely high biodiversity. The size and orientation of the ecoregion means that climatic conditions vary markedly – the north being wetter and having less seasonal variability compared to the south. The inaccessibility of the region has aided in its conservation.

4. The Madeira-Tapajós moist forests (fSE) are transected by the Transamazon Highway, and have high levels of urbanization and deforestation. There are characteristic liana (woody vine) forests with a lower (< 25 m) and more open canopy than the typical humid terra firme forests.

5. The Juruá-Perez moist forests (fC) are largely intact forests in the low Amazon Basin. The canopy can reach up to 30 m, with some patches of open canopy.

6. The seasonally flooded forest, Marajó várzea (ff), is located at the mouth of the Amazon River. The vegetation is dominated by palms, and shorter than surrounding forests. It has areas with tidal flows from the Atlantic Ocean, as well as seasonally and permanently inundated forests. The annual seasonal flooding occurs during the peak precipitation period between January-May (Camarão et al., 2002).

Three savanna ecoregions are also considered in this study:

1. The Cerrado (sC) borders the Amazon biome to the southeast. It occupies an area of 2 million $km^2$ in the Brazilian Central Plateau and is the second most extensive biome in South America (Oliveira et al., 2005). The vegetation cover varies from closed tree canopy to grasslands with low shrubs only (Eiten, 1972).

2. The Guianan savanna (sG) consists of forest patches encircled by extensive grasslands and shrub formations. The area is more susceptible to vegetation fires than typical humid moist forest environments and the dry season lasts from December-March.

3. The Beni savanna (sB) is a wetland region with riverine gallery forests and small forest islands. The landscape is dominated by the palm species Attalea princeps (Hordijk et al., 2019). Seasonal flooding occurs in up to half the region for 4 to 9 months, peaking in March-April (Hamilton et al., 2004).

Three Köppen-Geiger Climate Classes (KGCC) cover most of the study region (Fig. 2). The evergreen forest regions are classified as Af (tropical fully humid) or Am (tropical monsoonal), and the savanna regions have Aw (tropical winter dry) climate (Bradley et al., 2011). The annual precipitation in the forests can exceed 2000-3000 mm, with less than 100 mm rainfall for up to three months in the year. The savannas have a wet season extending for 5-8 months, with an annual total of 1000-2000 mm (Bradley et al., 2011). Net radiation peaks in the winter months, due to the absence of cloud cover in the dry season (Liu et al., 2018). Two major droughts occurred in the region during the study period, in 2010 and 2015 (Jiménez-Muñoz et al., 2016; Marengo et al., 2011) and are of particular interest in this study.

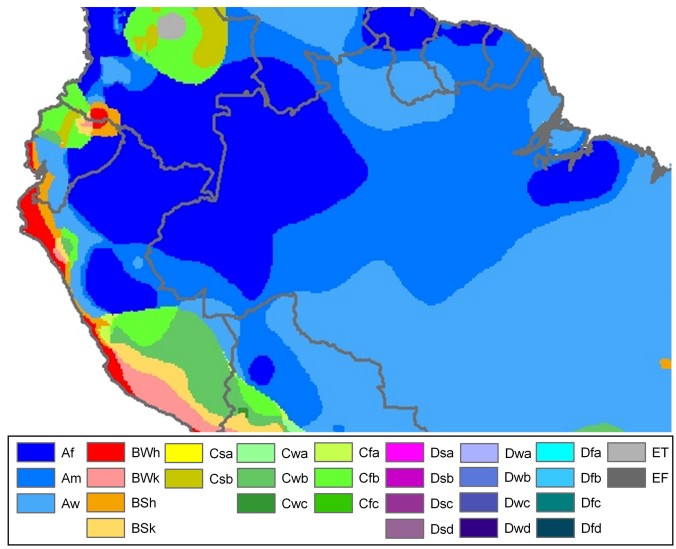

**Figure 2.** Köppen-Geiger climate zones in the study area (Source: (Peel et al., 2007)).

## 2.2 ASCAT data

The Advanced Scatterometer (ASCAT) data were processed using the same procedure as Steele-Dunne et al. (2019). Metop-A ASCAT SZR Level 1b Fundamental Climate Data Record, resampled at a 12.5 km swath grid, were obtained from the EU-METSAT Data Centre for the period 2007 to 2016. Following the procedure described by Naeimi et al. (2009), the backscatter observations were resampled to a fixed Earth grid using a Hamming window function. The slope and curvature were calculated from the ASCAT backscatter observations using the method introduced by Metzler (2013) and described by Hahn et al. (2017). The ASCAT instrument on-board the Metop satellites has three antennas on either side, oriented at $45°$ (fore), $90°$ (mid) and $135°$ (aft) to the satellite track. As a result, three independent measurements of each location on the surface are obtained almost instantaneously. These so-called "backscatter triplets" (Hahn et al., 2017) are used to calculate an instantaneous backscatter slope, also known as the "local slope":

$$\sigma'(\frac{\theta_{mid} - \theta_{a/f}}{2}) = \frac{\sigma^{\circ}_{mid}(\theta_{mid}) - \sigma^{\circ}_{a/f}(\theta_{a/f})}{\theta_{mid} - (\theta_{a/f})} \qquad [dB/deg] \tag{2}$$

where *mid*, *a* and *f* indicate the backscatter measurements from the mid-, aft- and fore-beams respectively. Following the approach of Metzler (2013), an Epanechnikov kernel (with width $\lambda=21$) is used to weight the local slope values by their temporal distance from a given day of interest. This yields an estimate of slope and curvature for a given day, based on all local slope values within a 42-day window. For a more detailed description of their derivation, the reader is referred to Steele-Dunne et al. (2019).

(Anderson et al., 2011) showed a calibration accuracy of Metop ASCAT backscatter of 0.15 - 0.25 dB. However, radiometric accuracy is expected to be better (i.e. less noisy) over stable, homogeneous targets (e.g. evergreen rainforest). To reduce noise,

the backscatter data is averaged in space (over the ecoregions of interest) and/or time (to monthly or dekadal intervals). The
number of grid points averaged is provided in Tables S1 and S2. Data are available every 1-2 days (Wagner et al., 2013).
Observations from the descending and ascending overpasses are unlikely to occur on the same day. Hence, the $\sigma^\circ_{40}$ data were
aggregated into 10-day intervals (dekads). Unless otherwise indicated, the analysis uses data from the descending pass only ($\sim$
10 am). Diurnal differences refer to the values from the descending overpass ($\sim$ 10 am) minus the values from the ascending
overpass ($\sim$ 10 pm).

## 2.3  Water Dynamics data

Downwelling shortwave radiation at the surface and specific humidity were obtained from the Princeton meteorological dataset
(Sheffield et al., 2006). These data have a $0.5° \times 0.5°$ daily resolution. Precipitation data were obtained from the Global
Precipitation Climatology Product (GPCP) Precipitation Level 3 Monthly 0.5-Degree V3.0 beta dataset (Huffman et al., 2009).
Precipitation, radiation and humidity are hypothesized to be the main atmospheric forcing for vegetation activity in the Amazon
((Nemani et al., 2003)). Therefore, these three forcings are compared to slope and curvature. As they are on similar temporal
and spatial scales, quantitative comparisons are performed. Data from the Gravity Recovery and Climate Experiment (GRACE)
mission were used to provide insight into terrestrial water storage variations (Landerer and Swenson, 2012; Swenson and Wahr,
2006). Here, we used the equivalent water thickness (EWT) from the GRACE Tellus dataset which is available at $1° \times 1°$,
monthly resolution from the NASA JPL Physical Oceanography Distributed Active Archive Center (PO DAAC). These data
give the relative change in EWT with respect to a baseline, the method of calculation for which is explained by Wahr et al.
(1998). These data provide information on fluctuations in EWT on monthly to inter-annual timescales. Note that EWT includes
variations in all terrestrial water storage terms including groundwater and surface water, in addition to the variables of interest
in this paper, namely soil moisture and vegetation. Furthermore, EWT is based on monthly data with a spatial resolution of
hundreds of kilometers. Statistical comparisons between the EWT and ASCAT would be strongly influenced by the sensitivity
of EWT to ground- and surface water and by artefacts of the difference in spatial and temporal scale between the two products.
Therefore, EWT is only qualitatively compared to backscatter, which is affected by soil moisture and vegetation.
Seasonal cycles were determined for precipitation, radiation, humidity, and EWT by averaging data from the entire study
duration. Anomalies in precipitation during the drought years were also calculated (as drought year values minus climatology)
to provide an indicator of the water stress against which to compare the backscatter, slope and curvature anomalies.

## 3  Results and Discussion

### 3.1  Seasonal Climatology

Figure 3 shows the mean and range of normalized backscatter ($\sigma^\circ_{40}$), slope and curvature for the study period (2007-16). In
general, the spatial patterns in the mean and range of all three quantities reflect the spatial patterns in land cover expected
from Fig. 1. It is striking that even the influence of the riverine network on the vegetation cover is discernible in the maps,
particularly that of the mean backscatter (Fig. 3(a)). Striping effects are visible in several of the maps, particularly that of the
range in curvature (Fig. 3(f)). This is due to the backscatter observations at the swath edges being available only at very high
or very low incidence angles, which skews the calculation of the slope and curvature. This effect is particularly noticeable in
forest regions where the natural dynamic range in both quantities is limited.
Mean backscatter is highest, with the least variability, in the evergreen forest regions (Fig. S1). Mean backscatter is 2-2.5 dB
lower in the savanna areas, but the range is up to 3 dB, compared to just 0.5 to 1 dB in the forest. The stability of the forest
is also apparent in the maps of slope and curvature. Though there is some variability among the forest ecoregions, the most
striking differences in slope and curvature are between the forest and savanna areas. Limited structural and water content
changes in the forest canopy result in a limited range of slope and curvature values in the forest ecoregions. The range of
both slope and curvature are highest in the Cerrado areas (Fig. S1). One interesting feature is the difference in mean slope
between the Guianan savanna (sG) in the north and the Cerrado (sC) region in the south. The Guianan savanna, with sparse
vegetation, has low mean slope values. The Cerrado, on the other hand, shows mean values higher than the evergreen forests.
This is unexpected since slope is generally considered a measure of "vegetation density", and the evergreen forests are much
denser than savannas. This will be discussed in detail in Sect. 3.1.1. Seasonal flooding of the Marajó várzea (the seasonally
flooded forest) and Beni savannas ensure that both ecoregions have strong seasonal cycles in all three quantities. These will be
discussed separately in Sect. 3.1.2.
The mean seasonal cycles in backscatter for all ecoregions of interest are compared in Fig. 4 (a). This highlights the contrast
between the very stable evergreen forest regions and the flooded forest and savanna areas. The mean backscatter value is
high, with limited seasonal amplitude in the evergreen forest regions. Backscatter variations are so limited in these areas that
they have long been used as calibration targets for spaceborne radar (Birrer et al., 1982; Kennett and Li, 1989; Frison and
Mougin, 1996; Hawkins et al., 2000). In contrast, backscatter is generally low, but also exhibits strong seasonal variations in
the flooded forest and savanna areas. Figures 4 (b-f) show the seasonal variation in backscatter split out for five ecoregions
of interest, against the corresponding climatologies of precipitation and EWT. As the evergreen forest ecoregions showed
very similar climatologies, only the Jurua-Purus moist forest is shown as a separate plot. In all of the ecoregions, the maximum
backscatter occurs during the wet season, and a decrease in backscatter is observed during the dry season, though the amplitude
of the variations is obviously much smaller in the forest ecoregions. In each ecoregion, there is clear agreement between
the seasonality of EWT and backscatter. This indicates that backscatter is influenced by moisture availability in terms of
total terrestrial water storage, which includes groundwater storage. It is noteworthy that this temporal consistency between
backscatter and EWT is apparent for both forest (fC in Fig. 4 (b)) and the Guianan Savanna (sG in 4 (e)) despite the contrast
between almost zero (0.25 dB) variability in backscatter in fC and the 2.5 dB seasonal cycle in sG. Figure S2 shows temporal
correlation between backscatter and precipitation is low for all ecoregions. A strong negative correlation and strong positive
correlation are found with radiation and humidity for lags between -2 and 2 months, indicating that backscatter is lowest during
drier periods with higher radiation and lower specific humidity.
Figure 5 (a) summarizes the mean seasonal cycle in the slope for the ecoregions of interest. The difference between ecore-
gions is more pronounced than for backscatter. The seasonal cycle for the evergreen forest ecoregions are similar in magnitude

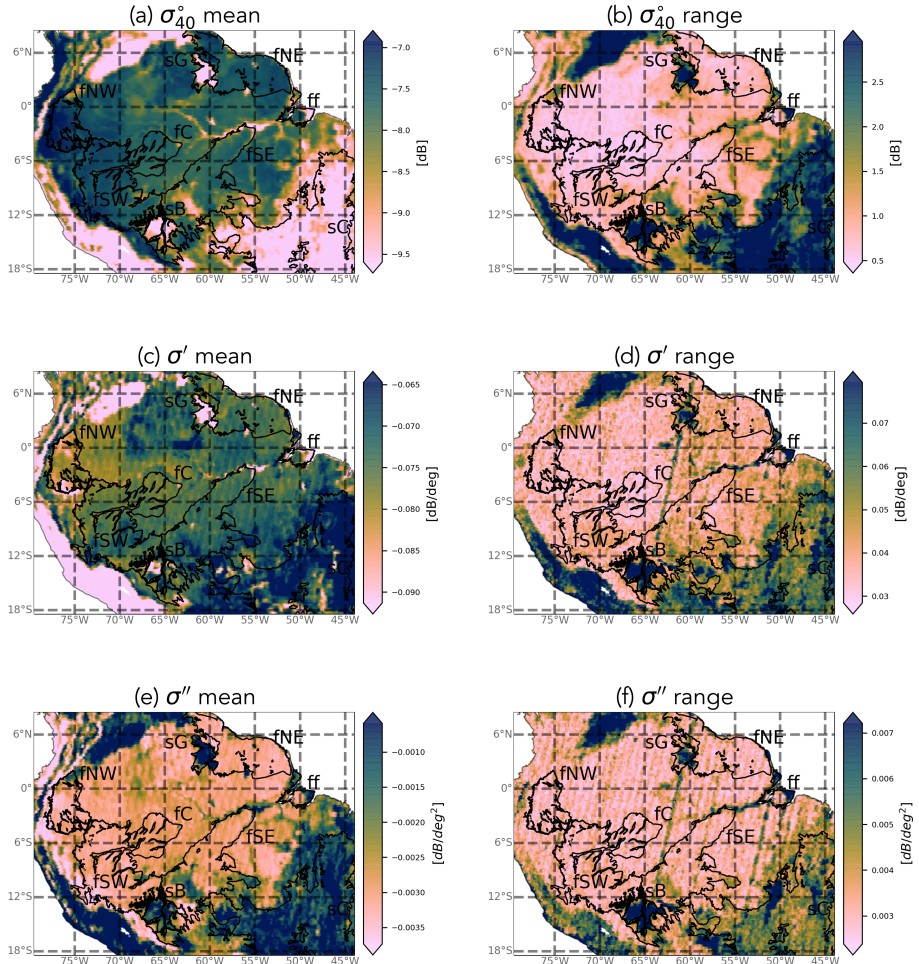

**Figure 3.** Mean and range of ASCAT normalized backscatter, slope and curvature in the study period (2007-16). Note that there are no data gaps, so white indicates that the quantity has a value equal to or less than the minimum value indicated on the colorbar.

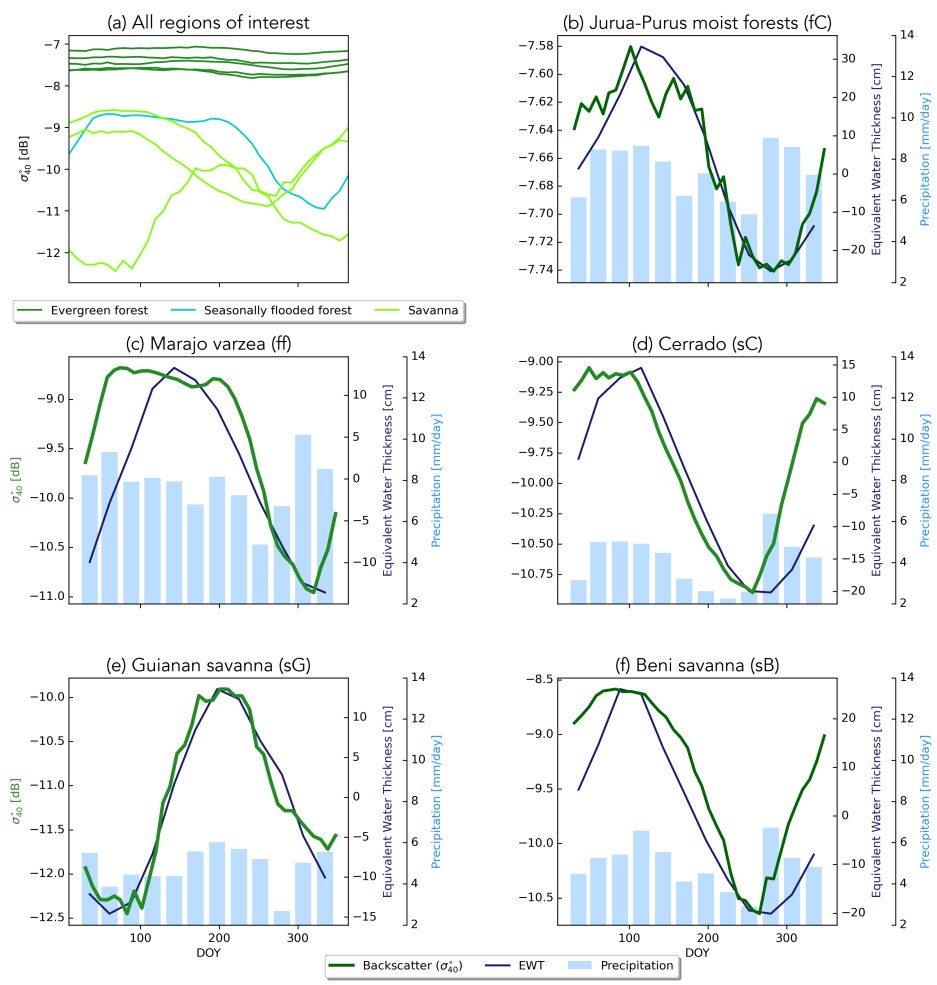

**Figure 4.** Climatologies of backscatter for all ecoregions; five evergreen forest (dark green), flooded forest (cyan) and three savanna (light green) (a). Plot (b) to (f) show climatology of backscatter (green line) with precipitation (bars) and EWT (blue line) per ecoregion. Note the different y-axes and that only the Jurua-Purus moist forest (fC) is shown as it is similar to the other evergreen forests.

but there are minor differences in the timing of the peak. The differences between the savanna regions are more pronounced than for backscatter. Significant differences can be seen in the mean slope value, as well as the amplitude and timing of the seasonal cycle of slope values for each ecoregion of interest.

In Fig. 5(b-f), the seasonal cycle of slope in each ecoregion is compared to the corresponding cycles of radiation, specific humidity and precipitation which drive photosynthetic activity in the region. Note again that only the Jurua-Purus moist forest is shown as a separate plot. Furthermore, Fig. **??** illustrates the temporal correlation between slope and precipitation, radiation and specific humidity. In the Jurua-Purus moist forests (Fig. 5(b)), the change is slope is one-tenth that observed in the other ecoregions. The variations in radiation and specific humidity are also very limited. Nonetheless, the seasonal cycle of the slope follows that of the radiation with a lag of about 30 days (Fig. **??**, R=0.75 at lag -1). This can be explained by the fact that the vegetation phenology in this tropical evergreen forest is driven by radiation (Romatschke and Houze Jr, 2013). The photosynthetic capacity depends on the available solar energy (Borchert et al., 2015). Energy availability drives transpiration and the accumulation of leafy biomass. This increases volume scattering from the canopy and therefore leads to an increase in the slope. Similar results were observed for the other forest ecoregions. In the Marajo varzea flooded forest (Fig. 5(c)), the variation in slope is much larger, and the seasonal cycle is clearly out of phase with that of the radiation. The seasonal variations in slope in this ecoregion are dominated by the influence of surface flooding rather than vegetation water content variations (Sect. 3.1.2).

In the Cerrado (Fig. 5(d)), there is a significant variation in specific humidity, and radiation as well as a strong seasonal cycle in precipitation. The peak in slope occurs during the driest time of year, when radiation is at a maximum and specific humidity and precipitation are at a minimum. Recall from Fig. 4, that this is also during the minimum EWT and backscatter period. This is also illustrated in Fig.**??** where strong negative correlations are found between slope and humidity. Correlations between slope and radiation are lower, and the highest correlation occurs at a lag of two months, i.e. slope leads radiation. Section 3.1.1 provides a detailed analysis of the vegetation types within the Cerrado ecoregion to better understand these variations. The slope values in the Guianan Savanna (Fig. 5(e)) are the lowest observed in all ecoregions, and also have the smallest variations among the non-forest cover types which are not strongly related to precipitation, radiation or specific humidity. This is consistent with the relatively low, but stable vegetation density associated with grasslands (Steele-Dunne et al., 2019). In the Beni Savanna (Fig. 5(f)), on the other hand, slope varies as much as in the Cerrado, and there is a very clear relationship between the slope and the atmospheric forcing data (Fig. 5 (f)). The maximum slope occurs at the peak of precipitation, EWT (from Fig. 4) and humidity. The minimum slope occurs during the dry season at the minimum in precipitation, humidity and EWT. This is consistent with the interpretation of slope as an indicator of vegetation density as the vegetation cover in this savanna changes dramatically in response to atmospheric forcing. This is also illustrated in Fig.**??**, where high correlations are observed between slope and humidity with small lags. The contrast in the seasonal cycles in slope in Fig. 5 reflect the diversity of the vegetation cover types in the ecoregions and their varied response to moisture supply and demand.

Figure 6 (a) shows the mean seasonal cycles in curvature for the regions of interest. The differences in the amplitudes of the seasonal cycles vary considerably among the regions. While the evergreen forests vary less than 0.0005 dB/deg$^2$, variations in the wetland regions (Beni savanna and Marajó várzea) are an order of magnitude larger. Aside from the Guianan savanna,

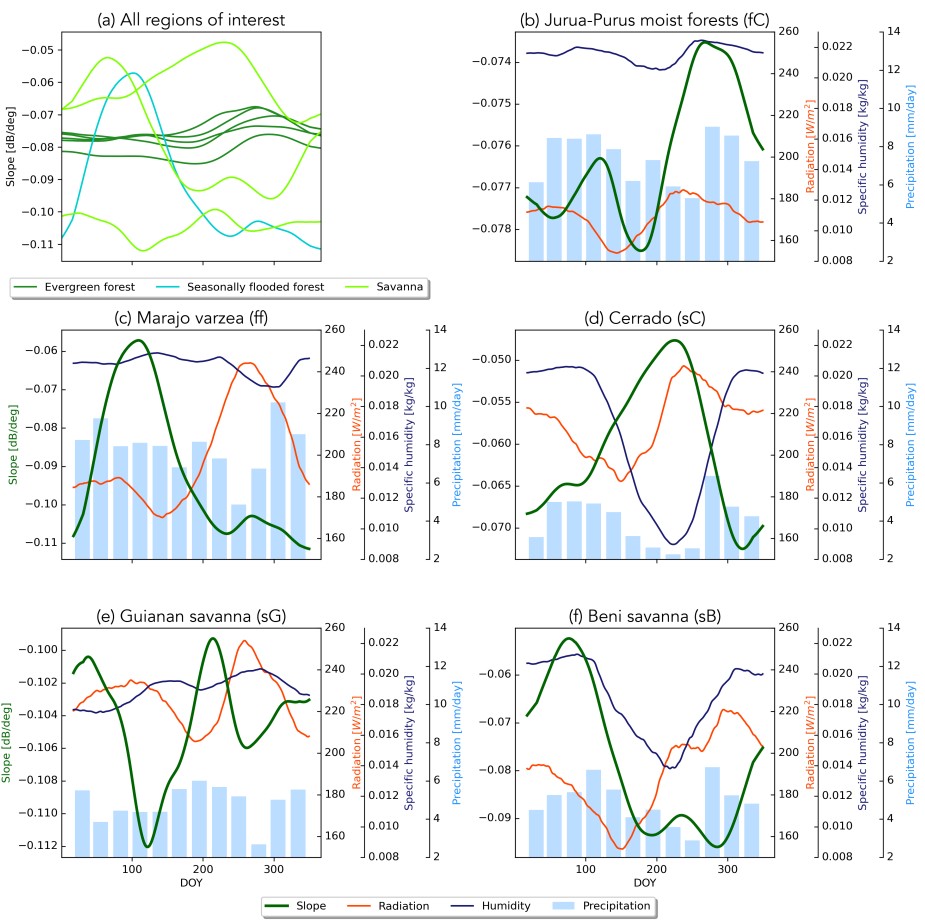

**Figure 5.** Climatologies of slope for all ecoregions; five evergreen forest (dark green), flooded forest (cyan) and three savanna (light green) (a). Plot (b) to (f) show climatology of slope (green line) with precipitation (bars) and specific humidity (blue line) and radiation (red line) per ecoregion. Note the different y-axes and that only the Jurua-Purus moist forest (fC) is shown as it is similar to the other evergreen forests.

the timing of the seasonal cycle is similar across all ecoregions. Previous research has suggested that curvature is related to vegetation phenology and structure (Steele-Dunne et al., 2019). Since the vegetation phenology in much of the forested region is radiation-driven, we hypothesize that the curvature seasonality is related to the radiation and evaporative demand. In the Amazon rainforest, Borchert et al. (2015) observed that leaf flushing and flowering in adult trees of numerous species coincided with the rise and decline of insolation. Wagner et al. (2016) made a similar observation about leaf flushing and rising insolation in July, and also noted that the litterfall peak occurs when evaporative demand is highest and can persist through the dry season. Figure 6 (b) shows that although the changes in curvature are very small in the rainforest, the peak occurs in July on the rising limb of the radiation data, and when the specific humidity is near its minimum. Figures 6 (b-f) and Fig. ?? show the strong correspondence between curvature and radiation (positive correlation at a lag of 2 months) and specific humidity (negative correlation with a lag of -1 month) and that the highest values of curvature generally correspond to lower humidity, higher solar radiation and lower precipitation. This suggests that higher values of curvature may be related to litterfall during periods of high evaporative demand. It is also noteworthy that the curvature values in the Guianan savanna (Fig. 6 (e) are positive for much of the year, consistent with the dominance of grass cover in this region.

### 3.1.1 Cerrado

As described in Section 3, the Cerrado shows a peak in slope, which indicates increased volume scattering, at a time of low precipitation and humidity, maximum radiation and low backscatter. To better understand these variations backscatter, slope and curvature are analyzed for the entire Cerrado region per land cover class. An overview of the number of used grid points per land cover can be found in (Supplementary) Table S2. Figure 7 provides a detailed map of the Copernicus Global Land Service Land Cover within the Cerrado region Buchhorn et al. (2020). The dominant cover types are herbaceous cover and shrubland, with patches of cropland and forest. Figures 8 and 9 show the spatial patterns and boxplot per land cover type of mean, maximum and the DOY of the maximum for backscatter, slope and curvature. The mean backscatter varies between -13 and -7 dB and is highest for forest regions and lowest for croplands. The DOY for the maximum backscatter varies with latitude, from December to January in the southern region to April in the northern region. As expected, the highest backscatter corresponds with the months of highest precipitation and EWT, the minimum in backscatter corresponds with the months of lowest moisture availability (Fig. 4). The seasonal dynamics in backscatter are strongest in cropland. This may be related to the higher sensitivity to surface soil moisture in croplands and low backscatter may be related to dry surface soil conditions. The slope mean and maximum values show a decrease from shrubs to herbaceous to cropland, decreasing with vegetation density as expected. Forests are characterised by high mean and maximum slope values. The seasonal dynamics and DOY of the maximum slope vary strongly with land cover type. In croplands, the maximum slope, i.e. where volume scattering is highest, occurs between DOY 340-150. This corresponds to the highest precipitation and EWT, indicating increased vegetation density. In natural vegetation, such as herbaceous cover, shrubs and forests, the highest slope occurs between day 200 and 300 and coincides with the minimum in precipitation and EWT but with maximum radiation (Fig. 5). This is illustrated in Figure 10, where slope and radiation dynamics for different land cover classes are depicted. To exclude confounding effects due to heterogeneous land cover within ASCAT pixels, we used only pixels with a dominant land cover fraction of $> 80\%$. The slope

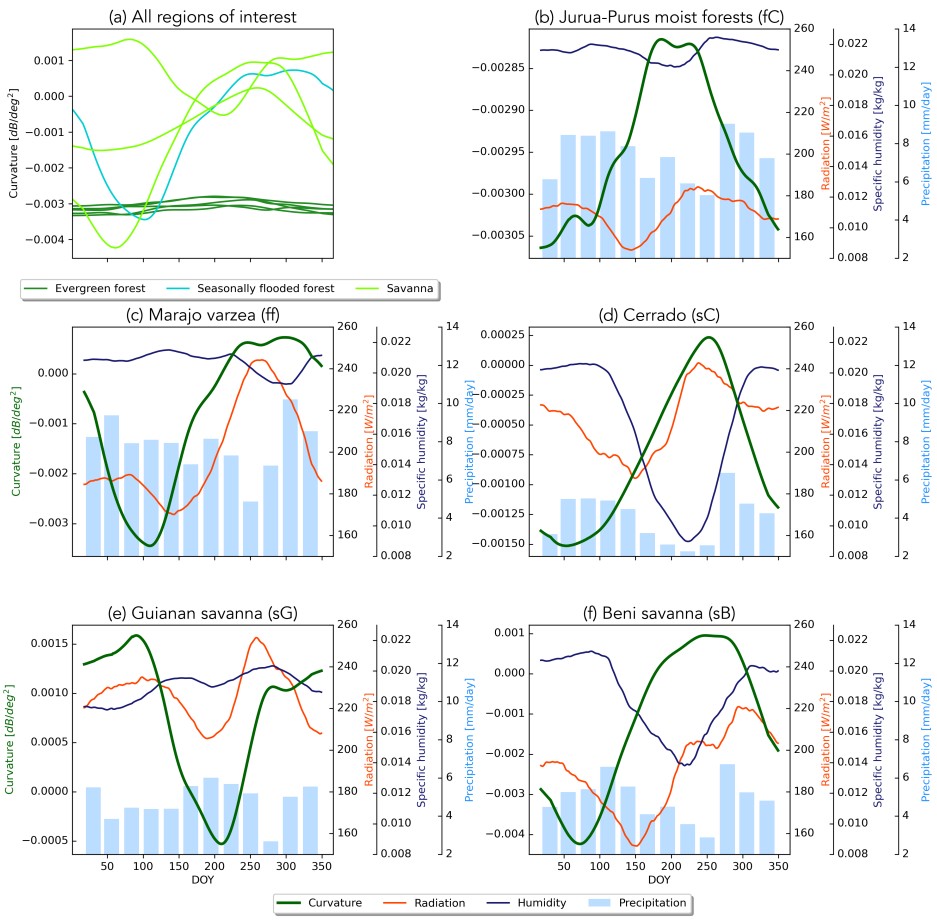

**Figure 6.** Climatologies of curvature for all ecoregions; five evergreen forest (dark green), flooded forest (cyan) and three savanna (light green) (a). Plot (b) to (f) show climatology of curvature (green line) with precipitation (bars) and specific humidity (blue line) and radiation (red line) per ecoregion. Note the different y-axes and that only the Jurua-Purus moist forest (fC) is shown as it is similar to the other evergreen forests.

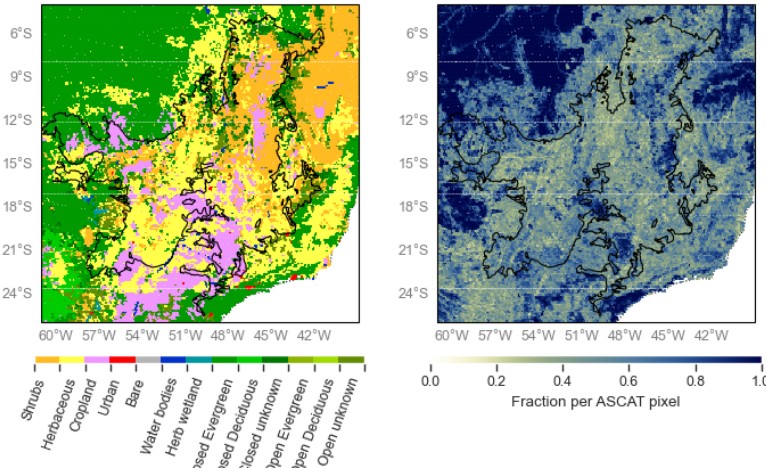

**Figure 7.** Dominant land cover type (left) and fraction (right) derived from the Copernicus Global Land Service Land Cover (2015) for the Cerrado region Buchhorn et al. (2020)

dynamics in cropland are following the precipitation dynamics and have their peak during the wet season. Herbaceous cover shows two peaks in slope, one coinciding with the wet season at the beginning of the year, and a higher peak coinciding with the dry season and maximum in radiation. The increase in slope coincides with the onset of the increase in radiation. In shrubs and forests, slope starts to increase after the wet season, but before the increase in radiation (Fig. 10). This counter-intuitive behavior of the slope over natural vegetation can be explained by the variability in limiting factors to vegetation activity. Within the Cerrado region, vegetation can be moisture limited or energy limited (Nemani et al., 2003), depending on location and land cover type. Contrary to crops, natural vegetation types such as herbaceous vegetation, shrublands and forests have deeper root systems and they can tap into deeper water reservoirs. This enables them to increase photosynthesis and leaf development slightly before or at the onset of increasing radiation even though precipitation is at its minimum. The increase in vegetation activity will lead to increased volume scattering and a flatter backscatter over all incidence angle and subsequent higher slope. Chave et al. (2010) found that, among the tropical forest types in South America, the highest seasonality in litterfall was observed in "low" stature forests, such as those found in the Cerrado. They also cite Wright and Van Schaik (1994) to argue that seasonality of solar radiation rather than precipitation may be the most important trigger for leaf flushing and leaf abscission. Croplands and herbaceous vegetation show positive curvatures, whereas forests are characterised by negative curvatures with the maximum values occurring between DOY 200 and 300 across the Cerrado. The positive curvature for crops and herbaceous vegetation can be explained by the vertical structure of the vegetation.

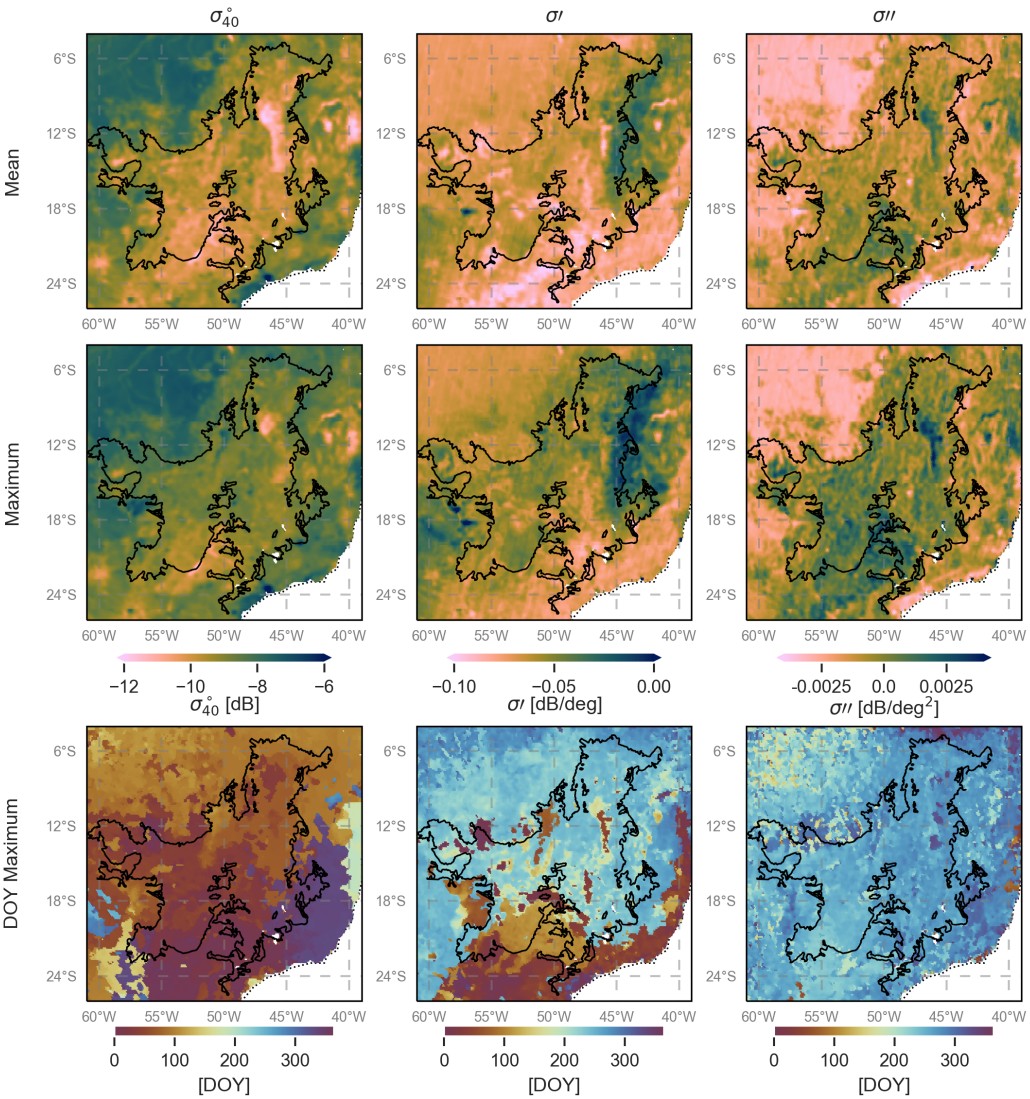

**Figure 8.** Mean, maximum and day of year of maximum for backscatter, slope and curvature over the Cerrado.

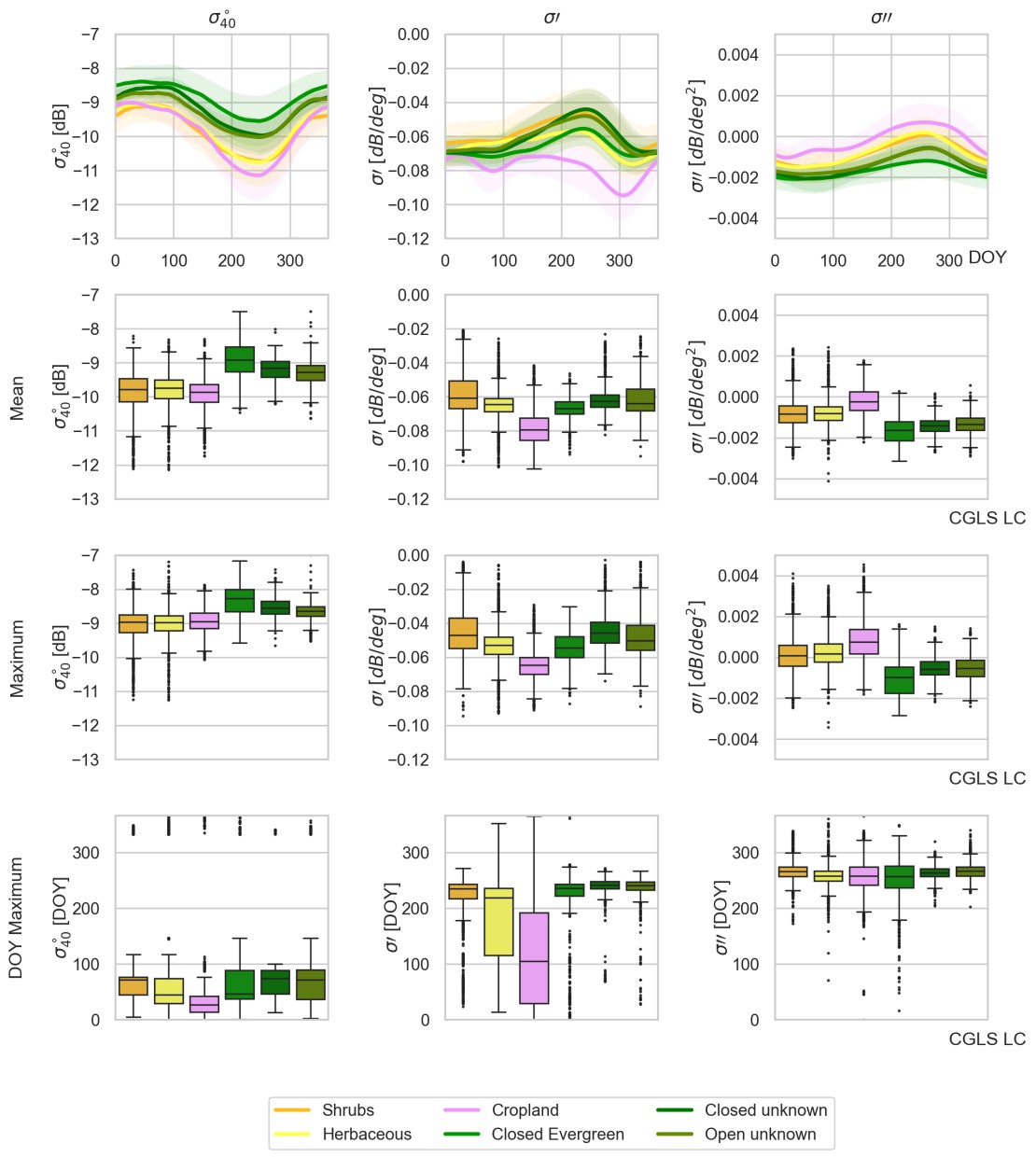

**Figure 9.** Time series averaged per land cover class and boxplots of mean, maximum and day of year of maximum for backscatter, slope and curvature over the Cerrado.

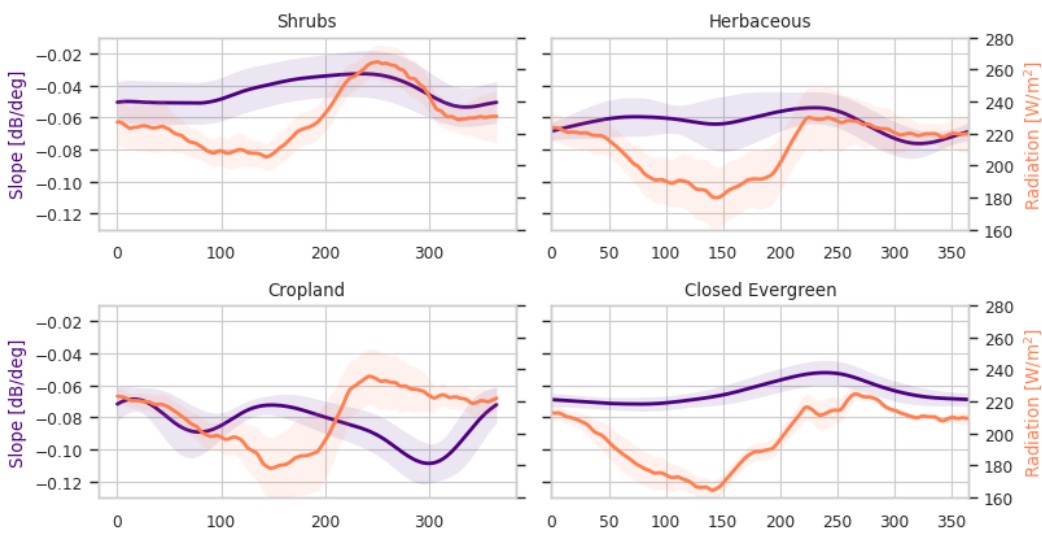

**Figure 10.** Seasonal cycle of slope and radiation per land cover class in the Cerrado region. Only ASCAT pixels in which the fraction of the dominant land cover type exceeds 80% are included.

### 3.1.2  Seasonal Flooding

Fig. 11 shows the striking effect of seasonal flooding on the incidence angle dependence of backscatter. This relationship was obtained using Equation 1 for a reference angle of $40°$ with the climatological mean values of $\sigma_{40}°$, slope and curvature for several days during the year. The flooded period is indicated in shades of blue. First, note that $\sigma_{40}°$ is around 2 dB higher during the seasonal flooding. Under forest/woody vegetation, this is due to a combination of double bounce scattering between the surface and trunks, and multi-path scattering between the surface and the vegetation (Townsend, 2002).

Recall from Fig. 5 that the slope is slightly higher during this period as this multiple scattering is apparently slightly less sensitive to incidence angle than scattering from the vegetation during non-flooded period. However, the most noteworthy difference is in the curvature. In both ecoregions, the curvature changes considerably and even changes sign during the flooded period. This illustrates that the curvature includes useful information on changes in the scattering mechanisms, which are related to physical changes at the land surface.

### 3.2  Diurnal Differences

Figure 12 shows the mean diurnal differences for backscatter and EWT in the study area for alternate months in the year, where positive values indicate that values are higher during the descending (10 am) overpass than those from the ascending (10 pm) overpass. The diurnal differences in backscatter are generally very small, with maximum values less than 0.15 dB. Although this is unquestionably close to the limits of the ASCAT sensor in terms of radiometric accuracy, these results are based on

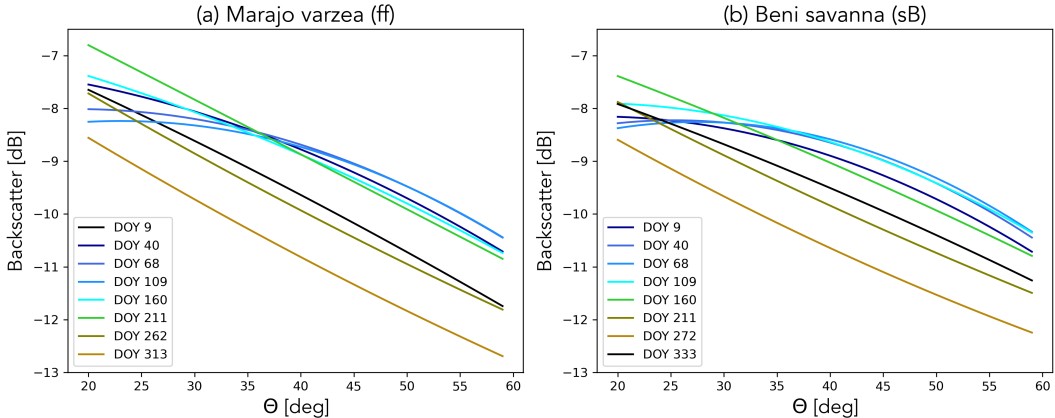

**Figure 11.** Averaged backscatter as a function of incidence angle for several dates in the Marajo varzea (a) and Beni Savanna (b) ecoregions.

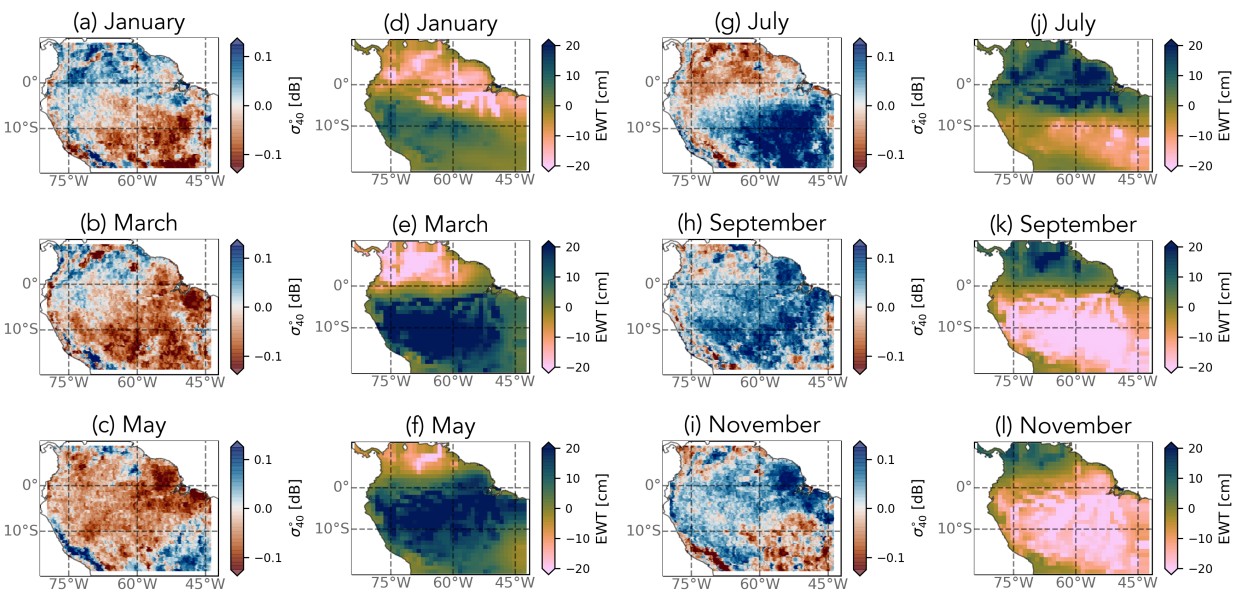

**Figure 12.** Maps of monthly mean diurnal differences in $\sigma_{40}^\circ$ (morning minus evening passes) and monthly mean Equivalent Water Thickness (EWT) from GRACE for different months in the year.

monthly averages, which is expected to reduce noise. Furthermore, there is a clear seasonal variation, broadly following that of EWT, demonstrating that patterns are likely not a result of noise.

For most of the domain, especially the evergreen forests, high values in EWT coincide with negative diurnal differences in backscatter and vice versa. During periods of maximum EWT, the backscatter is higher in the evening than in the morning.

This is consistent with the finding that precipitation in tropical South America (since it is generally produced by convective
systems) predominantly occurs in the late afternoons and evenings (Romatschke and Houze Jr, 2013). Hence, these higher
backscatter values are due to the canopy being wetter in the evening.
During the drier periods (e.g. September (h, k) and November (i,l) in the south of the study area), backscatter is higher at
10 am than at 10 pm, consistent with the loss of moisture through transpiration during the day. In a light-limited evergreen forest
such as the Amazon (rather than a water-limited forest), the canopy photosynthetic capacity seasonality is driven by radiation
(Wagner et al., 2016). When the plants are phenologically active, they lose water during the daytime through transpiration
resulting in lower evening backscatter values. The results in Fig. 12 are consistent with the findings of previous studies by
Frolking et al. (2011) and Friesen et al. (2012) who also found the morning backscatter over Amazonia to be higher (on
average) than the evening values due to higher water content in the vegetation. In the areas surrounding the evergreen forests,
the patterns can be less straight-forward. Note, for example, that the diurnal difference in $\sigma_{40}^\circ$ in the Guianan savanna (sG)
consistently has the opposite sign to that of the surrounding forest.
In Fig. 13, the seasonal cycle of the diurnal difference in $\sigma_{40}^\circ$ is compared to those of the radiation, precipitation and EWT
for each of the ecoregions of interest. Figure 13(a) is indicative of the seasonal variations observed across the evergreen forest
ecoregions. Note that the diurnal differences are very small (< 0.06 dB). Recall from Fig. 4, 5 and 6 that the backscatter, slope
and curvature in these evergreen forests was essentially stable throughout the year, so even this small diurnal difference is
noteworthy given the limited seasonal variation. As mentioned in the discussion of Fig. 12, evening values are higher than
morning values during the EWT maximum and vice versa. Diurnal differences are larger in the Marajo varzea (Fig. 13(b)), but
interpreting their seasonal variation is complicated by the seasonal inundation. In the Cerrado and Beni Savanna ecoregions
(Fig. 13(c) and (e)), the diurnal differences in backscatter are almost twice as large as those observed in the evergreen forest
regions. Morning values are up to 0.1 dB higher than evening values during the dry season due to loss of plant moisture during
the day. Similar to the forest regions, evening backscatter values are higher during the rainy season. The Guianan savanna
(Fig. 13(d)) is quite distinct in that morning backscatter is up to 0.15 dB higher than evening backscatter during the EWT
and backscatter peak. One possible explanation for this unusual seasonal cycle is that it is related to a change in the relative
dominance of the forests and grasslands in the backscatter signal. The transition from positive to negative curvature values
during the EWT peak indicate an increased contribution from tree patches and shrubs during the wetter period. The higher
backscatter in the morning may be due to water uptake in the trees during the night.

### 3.3 The 2010 and 2015 droughts

During the study period (2007-16), two major droughts occurred in Amazonia, in 2010 and 2015. Figure 14 shows the spatial
distribution of anomalies in $\sigma_{40}^\circ$, slope and curvature during the peak of the droughts from June to September 2010 and October
to December 2015. Two regions of interest are indicated in the maps, the savanna Cerrado (sC) ecoregion and Southwest
Amazon moist forests (fsW). The 2010 drought was most severe over southern and western Amazonia (Panisset et al., 2018).
The 2015 drought was considered a "record-breaking" event with stronger warming than that seen in previous events (Jiménez-

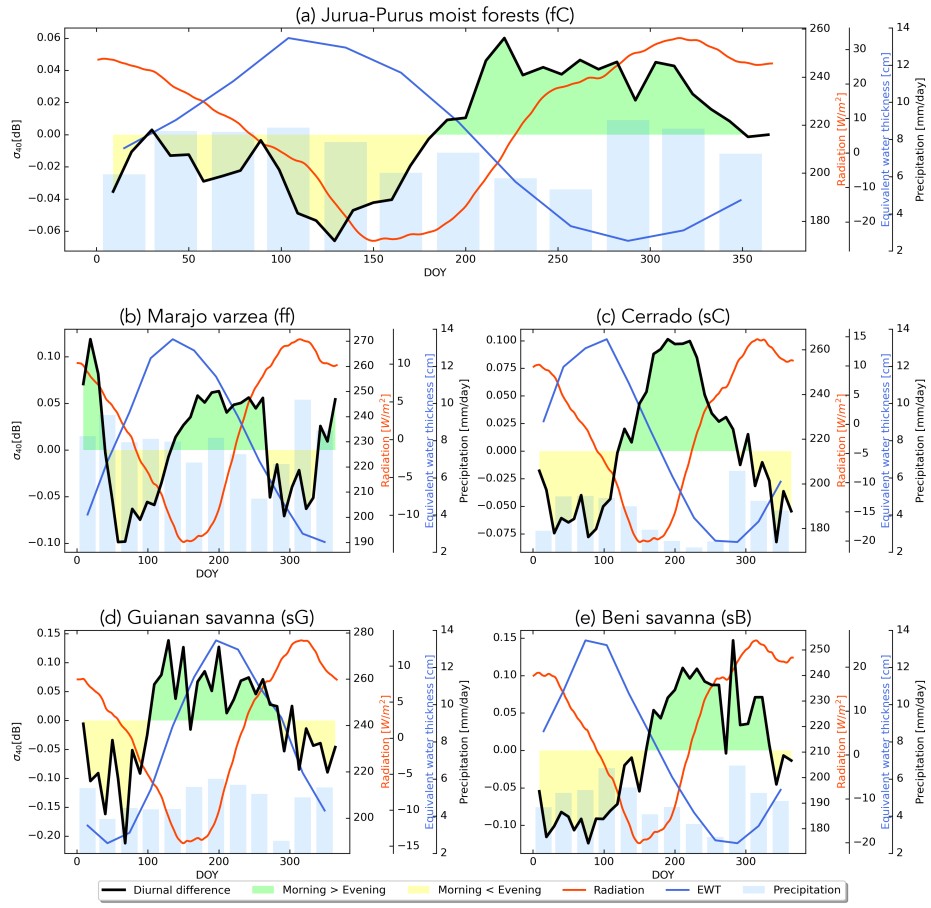

**Figure 13.** Seasonal cycle of diurnal difference in backscatter (black line), radiation (red line), EWT (blue line), and precipitation (bars) for different cover typres. Green (yellow) fill indicates days in which backscatter is higher (lower) in the morning than in the evening.

Muñoz et al., 2016). According to Panisset et al. (2018), there was a "pronounced lack of rainfall availability during late spring and early summer". The 2015 drought was more widespread than the event in 2010, and strongest in eastern Amazonia.

Negative anomalies are observed in $\sigma^\circ_{40}$, especially in the southern regions and in the Cerrado in 2010 and in eastern regions in 2015. Although anomalies are small over forests they are robust. Recall from section 2 that the backscatter noise level is low over evergreen forests and noise is further reduced through temporal aggregation. Note that the most eastern part of the Cerrado shows positive anomalies in 2010. The forests in fsW show minor negative anomalies (<0.1 dB) in $\sigma^\circ_{40}$ in 2010 and slightly stronger negative anomalies in 2015. Negative anomalies in backscatter from QSCAT were also observed during the 2005 drought (Saatchi et al. (2013); Frolking et al. (2017)). No significant spatial or temporal anomalies were observed in the diurnal differences in backscatter during the drought years. The slope and curvature do not show clear spatial patterns in anomalies during the 2010 drought, although the southern region shows slightly more positive anomalies. A clear positive

anomaly can be observed in the slope in eastern Amazonia in the 2015 drought. The curvature shows less clear patterns, although a striping pattern can be seen, likely related to swaths.

Figure 15 shows the time series of anomalies in backscatter, slope and curvature for the moist forests in fsW and the Cerrado region for the 2010 and 2015 drought. The backscatter, slope and curvature over the closed evergreen forest in fsW shows very little variation (both in time and space) during both droughts. A slight increase up to 0.002 dB/deg in slope can be observed during the peak of the 2015 drought. This demonstrates that the fsW forests are stable for satellite calibration. The Cerrado shows varying responses depending on land cover type and are therefore investigated further. Negative anomalies in $\sigma_{40}^{\circ}$ in cropland and herbaceous land cover can be seen during both droughts. Especially during the 2010 drought the croplands in the Cerrado are strongly affected, with a negative anomaly of >-1dB for some pixels. During the more extensive drought in 2015, $\sigma_{40}^{\circ}$ in forest is also affected and negative anomalies up to -1.5 dB are observed. The slope shows minor positive anomalies during the peak of the drought in 2010. In an analysis of drought impact on VOD over the forests in southern Amazonia, Liu et al. observed similar positive anomalies in VOD from May to August during the 2010 drought. Negative anomalies in VOD were only observed during later stages of the drought, from August to October. In 2010, negative slope anomalies in the Cerrado are observed from October on. During the 2015 drought strong positive anomalies in slope and curvature are present over the Cerrado especially in forests. Contrary to the drought of 2010, the peak of the 2015 drought occurs during the precipitation season in the Cerrado. Normally the precipitation season is characterised by lower radiation, and the positive anomalies in radiation during the drought might enhance vegetation growth. Note that the very small fluctuations in backscatter observed in Figures 14 and 15 may only be scientifically evaluated in rainforest regions, where the spatio-temporal backscatter dynamics (radiometric variations) are among the most stable in the world.

## 4 Conclusions

In this study, ASCAT backscatter, slope and curvature were analyzed in conjunction with meteorological data and terrestrial water storage from GRACE in the Amazon region. Previous results, limited to grasslands, had suggested that the slope and curvature contained useful information for monitoring vegetation water dynamics. However, the current study is the first to attempt to explain the spatial and temporal variations in slope and curvature in terms of seasonal variations in moisture availability and demand. Furthermore, it confirms that the conclusions of Steele-Dunne et al. (2019) can be extended to a wide range of cover types.

Results show that the unique viewing geometry of ASCAT provides valuable insight into vegetation water dynamics across a diverse range of ecoregions. The timing of the seasonal cycle of normalized backscatter was consistent with that of GRACE EWT, with the maximum (minimum) normalized backscatter coinciding with the maximum (minimum) EWT in all ecoregions. Spatial patterns in mean and range of slope reflect the ecoregions within the study area. The seasonal cycle in slope was found to follow the moisture availability and demand indicated by meteorological data and their influence on phenology. A detailed analysis per land cover type over the Cerrado demonstrated this. Slope dynamics were concurrent with precipitation in croplands and herbaceous cover, although herbaceous cover showed a second peak coinciding with the maximum in radiation.

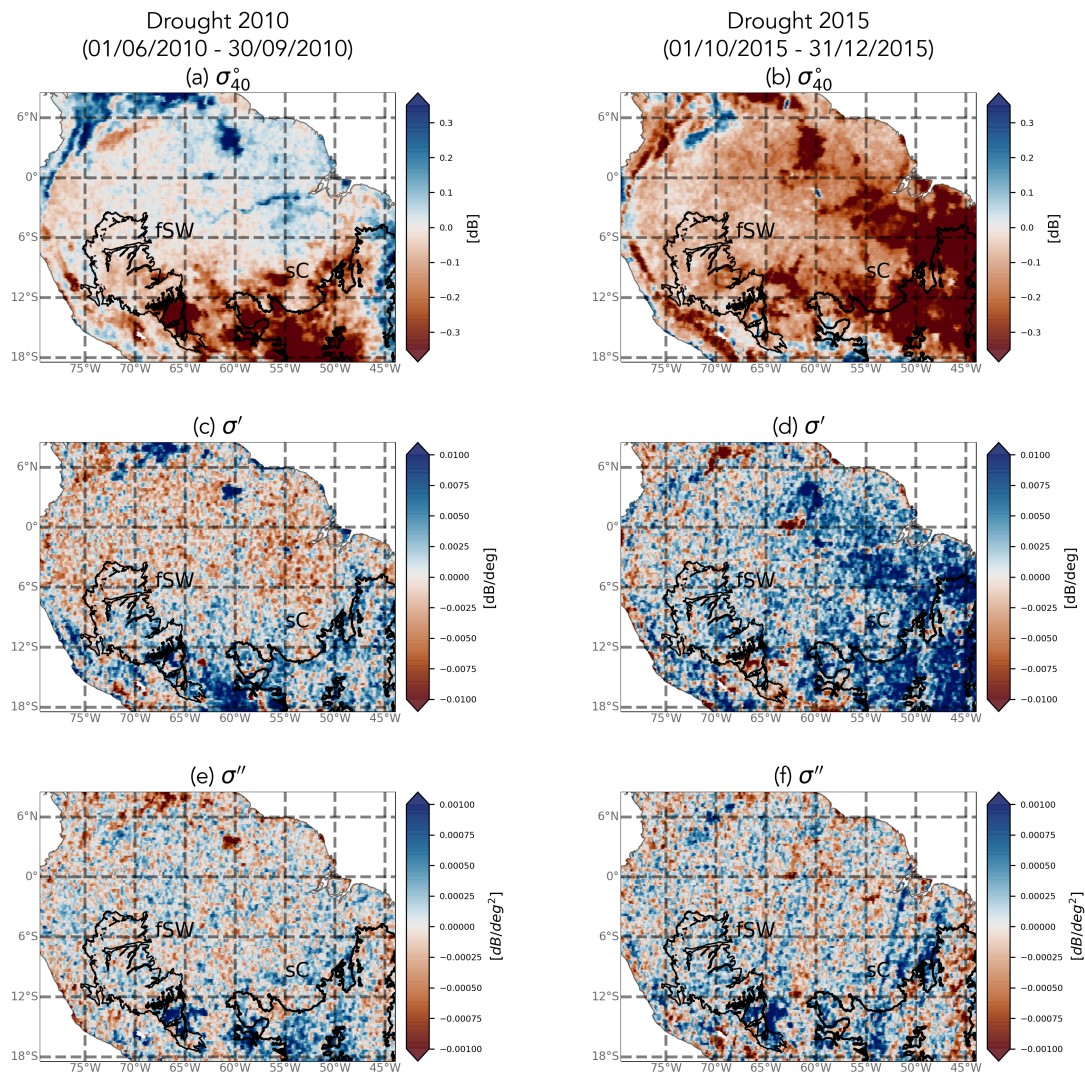

**Figure 14.** Spatial patterns in anomalies in backscatter, slope and curvature in response to the 2010 and 2015 droughts.

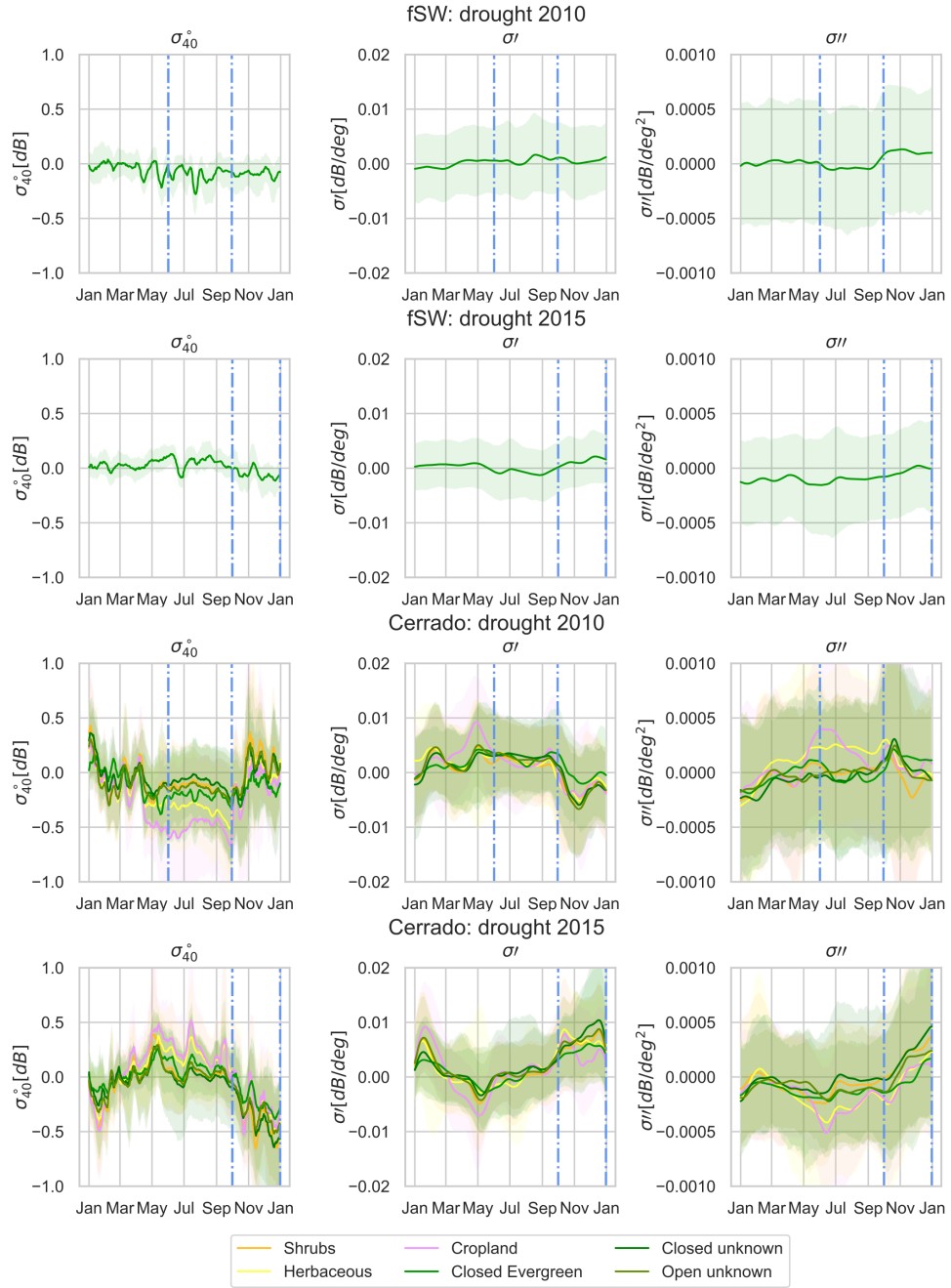

**Figure 15.** Time series of anomalies in backscatter, slope and curvature for moist forest (fsW) and Cerrado region. The shaded areas indicate the 5th and 95th percentile. The peak drought intervals (June-September 2010 and September-December 2015) are shown within dashed-dotted blue lines.

Slope dynamics in shrubs and forest corresponded with radiation, although the onset in increasing slope preceded the onset of increasing radiation. This may be due to leaf flushing, but it is difficult to draw a firmer conclusion given the limited availability of ground data (Chave et al., 2010). While the mechanism driving these variations in slope may not be immediately clear, it is important to note that there are open questions around the process of litterfall and its relation to precipitation and radiation in general. A recent study from Hashimoto et al. (2021) demonstrated that the temporal density of optical data from the Advanced Baseline Imager (ABI) onboard the Geostationary Operational Environmental Satellite 16 (GOES-16) yields unprecedented detail on the seasonality of NDVI and LAI in the evergreen Amazon forests. A comparison of ASCAT slope and curvature and ABI data may yield additional insight into the connection between slope, curvature and litterfall in the various ecoregions of our study area. Consistent with the findings of Steele-Dunne et al. (2019) in a study limited to grasslands, variations in curvature seem to be related to phenological change. The highest values of curvature coincide with periods of high evaporative demand (e.g. high radiation, lower humidity and lower precipitation). This suggests a link between curvature and phenological changes such as leaf flushing and litterfall. For example, the curvature peak in July in the rainforest occurs during rising insolation, and coincides with leaf flushing. Areas affected by seasonal flooding exhibited dramatic changes in both backscatter and curvature due to a suspected increase in multiple scattering between water on the surface and the vegetation.

Diurnal variations (i.e. the difference between morning and evening overpasses) were generally small, particularly in the evergreen forests. Nonetheless, their relation to the timing of precipitation highlights the importance of overpass time in using microwave observations for vegetation monitoring. Diurnal differences in backscatter during the dry season are dominated by transpiration losses. Long-term monitoring of these diurnal differences could provide insight into moisture availability and its influence on transpiration and vegetation functioning (Konings et al., 2021). Consistent with previous studies on the effect of drought on the backscatter signal over the Amazon forests (Frolking et al., 2011; Saatchi et al., 2013), a negative anomaly in backscatter was observed during the 2010 and 2015 drought, although this was minor for the moist forests, strong anomalies were observed in the Cerrado. The slope showed positive anomalies during the drought events in the Cerrado, similar to positive anomalies in VOD over forests observed by Liu et al. who attributed this to enhanced canopy growth due to increased radiation. Persistent positive anomalies in radiation were observed over the Cerrado, especially in 2015. The analysis confirms the confounding effects of mechanisms driving variation in slope in these regions.

For regions with non-closed-canopy conditions and significant soil contribution, the water sensitivity of the slope and curvature may be influenced, or even dominated by soil moisture dynamics (Greimeister-Pfeil et al., 2022). Furthermore, the various storage compartments (soil, vegetation) are linked by the soil-plant-atmosphere system. An improved physical understanding of the influence of both soil and vegetation on slope and curvature is essential. Future research should also include forward electro-magnetic modelling of multi-angular backscatter (i.e slope and curvature) to improve our understanding of how they relate to vegetation water and biomass variations as well as soil moisture.

The improved understanding of slope and curvature gained in this study is valuable in terms of our ability to use ASCAT for vegetation monitoring, and specifically for vegetation water dynamics. Slope and curvature may be influenced by the number and distribution of the scatterers, and their dielectric properties, all of which influence the optical depth i.e. the attenuation of the signal by the vegetation.Our improved understanding of the slope and curvature and how they are affected by vegetation

structure and water content, and interactions between the soil and vegetation is essential to improve our ability to interpret and optimally use VOD derived from ASCAT. Therefore, this research contributes directly to the continued development of the ASCAT VOD products. For example, it provides further insights in the VOD calculated from ASCAT by Vreugdenhil et al. (2016), where the main temporal dynamics stem from the slope and curvature. Furthermore, the fact that the slope and curvature themselves reveal different aspects of the vegetation response to the balance between moisture availability and demand means they are potentially useful low-level observables, i.e. they are obtained with minimal processing, and avoid the assumptions and simplifications required to retrieve geophysical variables. The results of this study suggest that their information content can be directly exploited to monitor vegetation water dynamics. The current study was performed over different land cover types, demonstrating the potential to study vegetation water dynamics with these observables over different regions. However this research also confirms the need for further research to overcome the limited understanding of the spatio-temporal dynamics of slope compared to environmental drivers and effects in structure of vegetation. A lot of our understanding of the incidence angle dependence of backscatter is based on experiments with tower-based or airborne radar systems conducted in the 1970s to 1990s (e.g. Ulaby (1975); Ferrazzoli et al. (1992)) to optimize the design of spaceborne radar systems. However, these experiments were generally focused on classification, soil moisture or LAI/biomass retrieval. Radar data were limited in space and/or time, and water dynamics (beyond soil moisture) were often not considered. Recent studies have focused on the relation between water dynamics in vegetation and tower-based radar backscatter response(e.g. (Vermunt et al., 2020; Khabbazan et al., 2022)), but not at the slope of the backscatter incidence angle relationship. So, in any first-order ground validation, we advocate the inclusion of incidence angle dependence. Nonetheless, field-based experimental campaigns have the disadvantage that they are very localized. Thus, studies like the one presented here, to explore ASCAT dynamic vegetation parameters and explain the variations in terms of modeled or observed geophysical variables are equally valuable and needed, because they allow us to study a wide range of cover and climate types and the impact of events such as drought. Based on this, and considering the planned SCA instrument on Metop-SG, incidence angle variations should be studied in more detail and be considered as a potentially valuable source of useful information. Ongoing research is focused on using data-driven and radiative transfer modeling approaches to investigate the sensitivity of slope and curvature to physical changes at the land surface including also different regions and cover types.

*Author contributions.* AP, SSD and MV were responsible for the conceptualization, methodology, formal analysis, investigation, visualization and writing (original draft preparation). SH provided resources (ASCAT data). RO contributed to the investigation. SSD and MV provided supervision. All authors contributed to writing (review and editing).

*Competing interests.* The authors declare that no competing interests are present.

459 *Acknowledgements.* Susan Steele-Dunne was supported by The Netherlands Organization for Scientific Research (NWO) User Support
460 Programme Space Research (Project ALWGO.2018.036 - 'A new perspective on global vegetation water dynamics from radar satellite
461 data') and NWO Vidi Grant 14126. Mariette Vreugdenhil was supported by ESA's Living Planet Fellowship SHRED (contract number
462 4000125441/18/I-NS).

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
