# Peer review of "The influence of vegetation water dynamics on the ASCAT backscatter-incidence angle relationship in the Amazon"

_Hydrology and Earth System Sciences, 2021_

## Author Comment (AC2)

**Reviewer comments 2:**

(https://hess.copernicus.org/preprints/hess-2021-406/#RC2)

**General Comment:**

The submitted manuscript presents the analysis of ASCAT time series data (backscatter, slope & curvature) over the greater Amazon region with regards to water dynamics and two drought events. Additional meteorological (e.g. precipitation from GPCP) and water dynamics (from EWT – GRACE) information are incorporated into the analyses for visual and chart comparison. The following comments & suggestions are raised concerning study setup, additional analyses and hopefully useful suggestions to improve the manuscript.

Thank you for your careful consideration of our manuscript and the constructive feedback.

**Major Comments:**

1. The study analyses are based on very small changes in backscatter (sometimes well below 0.1 dB in variation). This puts a massive demand on radiometric stability (and NESZ) of the ASCAT sensor. Please elaborate on this topic and include justifying statements. How far are these small backscatter variations showing significant and stable correlations to variations in environmental properties in the Amazonian vegetation? Is there a lower limit in sensitivity? The reviewer thinks it would be reasonable to define a lower limit.

   Wilson et al. (2010) mention that ASCAT was expected to have an accuracy of +/-0.3dB at 95% confidence level. A subsequent validation study by Anderson et al. (2011) showed a calibration accuracy of 0.15 - 0.25 dB. Therefore, changes on the order of 0.1dB are unquestionably close to the limits of the sensor. It is important to note, however, that the radiometric accuracy is expected to be better (i.e. less noisy) over stable, homogeneous targets (e.g. evergreen rainforest). Furthermore, the results presented here have been averaged in space or time, or both, which also reduces the noise. Consequently, it is reasonable to assume that the spatial and temporal patterns observed can be attributed to geophysical variability rather than observation error. These sentences will be included in the discussion.

2. The study shows mainly a chart/map comparison of the included observations, like in Figures 4, 5 & 6 However, there is no quantitative statistical analysis (tables) of the spatial correlations of the different observations. In addition, Figures 3,8, 12, 13 and 15 show interesting spatial patterns (maps) of the observables and their anomalies. But, geo-statistics (spatial statistics) and their analysis are not undertaken. It would be interesting to look at a spatial correlation map of different fields (e.g. EWT vs. backscatter/slope/curvature). Please add spatial correlation statistics (maps) to the analyses of the manuscript.

   We have done this. However, there were multiple issues we encountered and therefore decided not to put the maps in the manuscript. First, it is important to note that EWT is derived from GRACE, which has a spatial resolution on the order of hundreds of kilometers, and is based on measurements collected over a monthly period. Furthermore, EWT includes all contributions to terrestrial water storage including soil moisture, fresh biomass, but also groundwater. Spatial or temporal statistics between EWT and ASCAT observables would contain artefacts of this mismatch in temporal and spatial resolution, and sensing volume. Therefore, correlation statistics between ASCAT observables and EWT are of questionable value.

Precipitation, radiation and humidity are at a higher spatial resolution and correlation statistics between them and the ASCAT observables can provide additional insights. However, as there are phase differences between the observables and the environmental variables a spatial correlation map is not very informative. Therefore, we have performed a correlation analysis with different lag times between the observables and meteorological variables. The figure below illustrates the correlation coefficients averaged per region for different lag times between backscatter, slope and curvature with precipitation, radiation and humidity. We would add this figure as supplementary material (Fig. A1) to the manuscript and will include the following text in Section 3.

For backscatter:

"Temporal correlation between backscatter and precipitation is low for all ecoregions (Fig. A1). A strong negative correlation and strong positive correlation are found with radiation and humidity for lags between -2 and 2 months, indicating that backscatter is lowest during drier periods with higher radiation and lower specific humidity."

For slope:

"In the Jurua-Purus moist forests (Fig. 5(b)), the change is slope is one-tenth that observed in the other ecoregions. The variations in radiation and specific humidity are also very limited. Nonetheless, the seasonal cycle of the slope follows that of the radiation with a lag of about 30~days (Fig. A1, R=0.75 at lag -1). This can be explained by the fact that the vegetation phenology in this tropical evergreen forest is driven by radiation.

In the Cerrado (Fig. 5(d)), there is a significant variation in specific humidity, and radiation as well as a strong seasonal cycle in precipitation. The peak in slope occurs during the driest time of year, when radiation is at a maximum and specific humidity and precipitation are at a minimum. Recall from Fig. 4d, that this is also during the minimum EWT and backscatter period. This is also illustrated in Fig. A1 where strong negative correlations are found between slope and humidity. With radiation correlations are lower and highest correlation occurs at a lag of two months, i.e. slope precedes radiation. Section \ref{subsec:cerrado} provides a detailed analysis of the vegetation types within the Cerrado ecoregion to better understand these variations. The slope values in the Guianan Savanna (Fig. 5(e)) are the lowest observed in all ecoregions, and also have the smallest variations among the non-forest cover types which are not strongly related to precipitation, radiation or specific humidity (Fig. A1). This is consistent with the relatively low, but stable vegetation density associated with grasslands.

In the Beni Savanna (Fig.5(f)), on the other hand, slope varies as much as in the Cerrado, and there is a very clear relationship between the slope and the atmospheric forcing data (Fig. 5(f)). The maximum slope occurs at the peak of precipitation, EWT (from Fig.4) and humidity. The minimum slope occurs during the dry season at the minimum in precipitation, humidity and EWT. This is consistent with the interpretation of slope as an indicator of vegetation density as the vegetation cover in this savanna changes dramatically in response to atmospheric forcing. This is also illustrated in Fig. A1, where high correlations are observed between slope and humidity with small lags. The contrast in the seasonal cycles in slope in Fig. 5 reflect the diversity of the vegetation cover types in the ecoregions and their varied response to moisture supply and demand."

For curvature:

"Figures 6 (b-f) and Fig. A1 show the strong correspondence between curvature and radiation (positive correlation at a lag of 2 months) and specific humidity (negative correlation with a lag of -1 month) and that the highest values of curvature generally correspond to lower humidity, higher solar radiation and lower precipitation.

[Figure]

Figure 1A: Temporal correlation between ASCAT observations (normalized backscatter (a-c), slope (d-f) and curvature (g-i)) and precipitation (left column), radiation (middle column) and humidity (right column).

3.    There is a lack of direct validation as backscatter, slope and curvature are "low level observables" concerning water dynamics in plants and cannot directly serve as vegetation parameters: Anyhow is a validation somehow, even in a future setup, possible? Please elaborate or discuss how a first-order validation could be conducted, potentially in an add-on study within a controlled environment. An improved understanding of the slope and curvature and how they are affected by environmental factors, here water dynamics in the Amazon, is needed to fully exploit the potential of the method. Curvature and slope are no direct indicators of plant density, phenology and structure. This is hard to link directly. Can we have an easier link? The reviewer likes to foster more discussion and outlook kind of statements in the later sections of the manuscript showing how to overcome the limited understanding of the spatio-temporal dynamics of slope/curvature compared to the environmental ones.

We agree. A lot of our understanding of the incidence angle dependence of backscatter is based on experiments with tower-based or airborne radar systems conducted in the 80s and 90s to optimize the design of spaceborne radar systems. However, these experiments were generally focused on classification, soil moisture or biomass/LAI retrieval. Radar data were limited in space and/or time, and water dynamics (beyond soil moisture) were not considered. A recent study from Kim et al. (2015) provided some insight into factors influencing incidence angle dependence of L-band backscatter based on airborne radar data. In light of the increasing use of microwave data for

vegetation applications, and increasingly related to water dynamics, we believe there is an urgent need for ground-based microwave experiments to improve our fundamental understanding of the links between microwave observables and water dynamics, and how microwave satellite remote sensing can optimally be used to observe vegetation water dynamics. This is the main goal of our field-based experimental research in which we are actively engaged (see Khabbazan et al. 2022, Vermunt et al. 2021, Vermunt et al. (2020) etc.). That said, one of the limitations of field-based experimental campaigns is that they are very localized.  ASCAT provides more than 10 years of global, spaceborne radar data. In addition to ground-based experimental research, we think studies like the one presented here, to explore ASCAT dynamic vegetation parameters and explain the variations in terms of modeled or observed geophysical variables are equally valuable because they allow us to study a wide range of cover and climate types and the impact of events such as drought. Based on the results presented here, and the planned SCA instrument on Metop-SG, we would argue that incidence angle variations should be considered as a potentially valuable source of useful information. So, in any first-order ground validation, we would advocate the inclusion of incidence angle dependence.  We will include a summary of these arguments in the discussion section.

4. Another fundamental question is: How much are backscatter, slope and curvature correlated in space and time? How much can be simply explained by only backscatter? This may have been addressed before (maybe in Steele-Dunne et al., 2019), but a statement/paragraph would be beneficial to justify the analysis of the derivatives (slope, curvature). This could be also supported by EM modelling efforts.

The slope and curvature provide insight into the relative dominance of surface scattering versus volumetric and multiple scattering. This information is not contained in normalized backscatter alone. The relationship between backscatter, slope and curvature varies per cover type, as shown in the current study (compare and contrast the flooded forest to the cerrado and evergreen forest, for example). The following lines have been included in the introduction to make it clearer that the slope and curvature contain information that is not contained in normalized backscatter alone. These sentences will be inserted between the description of the TU Wien Soil Moisture Retrieval approach and the summary of findings from Steele-Dunne et al. (2019):

"In other words, slope and curvature are calculated and used to account for the influence of vegetation in the soil moisture retrieval. An increase in soil moisture results in an increase in backscatter at all incidence angles, while a change in the vegetation (due to growth cycle or water status) changes the sensitivity of backscatter to incidence angle, i.e. it results in a change in slope and curvature. So, the slope and curvature provide complementary information to the normalized backscatter."

5.     Another point to discuss: How far is the presented water dynamics analysis transferable from Amazonas to somewhere else, e.g. other climates/biomes and regions of the world? More explanation would help to shape the potentials and limitations of the approach in the discussion section.

 We  included a variety of ecoregions to illustrate that the relation between slope and curvature and the atmospheric forcing data varies according to land cover and climate. This also demonstrated the potential of using slope and curvature to analyze vegetation water dynamics in multiple cover types. This had been demonstrated by Steele-Dunne et al. (2019), who showed slope and curvature show useful information on vegetation water dynamics in a study limited to grasslands in the central US. Of course, we realize that the Amazonas is a very specific region, and that in other climates new challenges will arise. We will add the following to the conclusion:

"The current study was performed over different land cover types, demonstrating the potential to study vegetation water dynamics with these observables over different regions. Ongoing research is

focused on using data-driven and radiative transfer modeling approaches to investigate the sensitivity of slope and curvature to physical changes at the land surface including also different regions and cover types."

6. Concerning vegetation penetration one major point is when the C-band EM waves start to interact with anything but not vegetation, like soil under vegetation: How far are soil influences on the backscatter signal playing a role, especially for lower vegetated or dry areas (e.g. Cerado)? Please evaluate and discuss potential non-vegetation influences on the signal, like soil scattering. Is there a criterion or threshold-based approach to find and exclude regions and/or times when non-vegetation effects, like from soil, have a too distinct/significant influence?

Microwave interactions with vegetated surfaces are complex due to the variety, in terms of size and dielectric properties, of the vegetation constituents and the influence this has on the propagation of microwaves through the vegetation, interactions within the vegetation and interactions between the soil and vegetation. We always consider the total backscatter as a combination of contributions from a soil-vegetation continuum. The rationale behind using the slope, for example, is that an increase in slope is indicative of a transition in behaviour between predominantly surface scattering (from the soil) to predominantly volume or multiple scattering (from the vegetation) and vice versa. This tells us if the normalized backscatter variations are due to soil only, vegetation only, or some combination of both. Naturally, for areas or periods when slope is low, there may be a contribution of soil scattering to the total backscatter signal. This is also reflected in the correspondence between EWT and backscatter, since EWT is an aggregated signal including soil and vegetation water among other elements. Thus, we agree that in terms of backscatter dynamics there might be added-value in further analyzing this, however the main aim of this research was to investigate the backscatter incidence angle relationship.

Working with the slope values allows us to observe the physical process of fresh biomass change (either due to phenology or water status) through its effect on the backscatter incidence angle relationship. As this is, in itself, an indicator of which type of scattering contributes to the signal, and these are continuous processes, we do not see the added-value of introducing criteria or thresholds to categorize behaviour rather than use the information directly.

6.     There is a spatio-temporal scale gap as well as a sensing volume gap (C-band EM wave penetration vs. 3D gravity field dynamics) between GRACE EWT and ASCAT observations. Hence, the reviewer has doubts that (lines 181-183)"…in each ecoregion, there is clear agreement between the seasonality of EWT and backscatter. This indicates that backscatter is influenced by moisture availability in terms of total terrestrial water storage, which includes groundwater storage." This is a strong statement and "a clear agreement" is not really statistically quantified. Please add some statistical or more quantitative analysis for justification of this agreement. Moreover, please explain and/or discuss the scale gap and sensing volume gap of the two remote sensing observations.

There is an obvious scale and sensing volume mismatch between the two datasets. This is why we do not consider it sound to provide a statistical or quantitative comparison between the two. It would be dominated by artefacts of the difference in spatial and temporal scale between the two products. We have included the following text in Section 2.3 by way of explanation.

"Precipitation, radiation and humidity are hypothesized to be the main atmospheric forcing for vegetation activity in the Amazon (\citep{nemani_climate-driven_2003}). Therefore, these three forcings are compared to slope and curvature. As they are on similar temporal and spatial scale quantitative comparisons are performed."

" EWT includes variations in all terrestrial water storage terms including groundwater, soil moisture, vegetation, and surface water. Therefore, EWT is only qualitatively compared to backscatter, which is affected by soil moisture and vegetation. "

Minor Comments:

1. Lines 171-172:

"The Guianan savanna, with sparse vegetation, has low mean slope values. The Cerrado, on the other hand, shows mean values higher than the evergreen forests. This is unexpected since slope is generally considered a measure of "vegetation density", and the evergreen forests are much denser than savannas."

The forest/vegetation density that microwaves "see" can be twofold. Density can come from dry biomass/structure, which is dry matter based, or come from vegetation water, which is wet matter based. This comment may help to review the above-mentioned paragraph.

Based on this comment, we need to make it clearer that when we discuss "vegetation density", we are not referring to plants per area, but actually to above ground fresh biomass which is indeed a combination of dry biomass and vegetation water content. We will make this clearer when slope is first discussed in the Introduction section: "Slope is considered an indication of vegetation density, or above ground fresh biomass, which is a combination of dry biomass and vegetation water content."

2. Figures 5 & 6:

These Figures contain four y-axes and show an overview how the incorporated parameters/variables behave along time. This is a first overview along time. What is missing is a statistical evaluation of the temporal correlation.

Refer to our response to major comment 2, particularly the inclusion of an additional figure quantifying the temporal correlation.

3. Lines 258-263 and lines 351-352:

"This is due to multiple scattering between the water surface and the vegetation." In terms of scattering mechanism characterization: Should this be double bounce scattering (water-vegetation)? What kind of scattering mechanism could this be?

Under forest/woody vegetation, this is a combination of double bounce scattering between the surface and trunks, and multi-path scattering between the surface and the vegetation (Townsend (2002). This has been clarified in the revised text and the reference has been added.

4. Lines 261-263:

"...the curvature changes considerably and even changes sign during the flooded period. This illustrates that the curvature includes useful information on changes in the scattering mechanisms, which are related to physical changes at the land surface." Is there a way to link the curvature more directly to the physical variables? Could the authors try modeling or anything similar? It would be interesting to couple a forward model with the slope and curvature metric to investigate sensitivities and dependencies. Could references (if done) or an outlook statement (if not yet done) be included in the manuscript?

In other research, we have developed a data-driven approach to simulate the ASCAT observables based on land surface variables (Xu Shan et al., RSE (in review)). Alternatively, Radiative Transfer

Modelling could be used to simulate C-band backscatter as a function of incidence angle to quantify the sensitivity of slope and curvature to physical changes at the land surface. We have added the following statement to the conclusions and outlook:

"Ongoing research is focused on using data-driven and radiative transfer modeling approaches to investigate the sensitivity of slope and curvature to physical changes at the land surface."

5.    Figures 12 & 13:

Can Figures 12 and 13 be shown in a way that they are jointly together and their similarity or difference in pattern can be understood intuitively? Maybe an add-on figure might be an option or a replacement of Figs. 12 & 13. For most of the domain, especially the evergreen forests, high values in EWT coincide with negative diurnal differences in backscatter and vice versa. In the moment, the comparison of two 6-pannel figures (12 & 13) appears complicated.

We have combined the two Figures into a single one to facilitate visual comparison of EWT and diurnal difference in backscatter. See Figure 12 of the revised manuscript.

6.    Figure 14:

Figure 14(a) is indicative of the seasonal variations observed across the evergreen forest ecoregions. Note that the diurnal differences are very small (< 0.06 dB). These seem to be really small differences.

How about signal stability in terms of radiometric resolution? i.e. How noise-prone are these subtle differences? Please add an explanatory paragraph and some discussion about this point.

Refer to our response to major comment 2. The variability in backscatter from evergreen forest ecoregions is extremely limited. In fact, the Amazon rainforest has long been used as a calibration target for spaceborne radar systems (Birrer et al. (1982); Kennet and Li (1989); Frison and Mougin (1996); Hawkins et al. (2000)). This has been mentioned in the revised manuscript in the discussion of Figure 4.

7. Lines 294-296:

"One possible explanation for this unusual seasonal cycle *could be* that it is related to a change in the relative dominance of the forests and grasslands in the backscatter signal. The transition from positive to negative curvature values during the EWT peak *could* indicate an increased contribution from tree patches and shrubs during the wetter period." How can this be justified? This sounds quite speculative. Should the statement be softened?

It would be inappropriate to draw a stronger conclusion without a detailed modeling study, for which detailed ground data (structure, vegetation water content) would be needed. Hence, we were careful to use "could" to indicate that this is a possible explanation but not one that we are currently able to confirm.

8. Lines 331-332:

"However, the current study is the first to relate the spatial and temporal variations in slope and curvature to moisture availability and demand." Are we really seeing a relationship that is statistically significant? Please add more statistical (correlation) analyses to support the statement.

This sentence has been revised to read: "However, the current study is the first to attempt to explain the spatial and temporal variations in slope and curvature in terms of seasonal variations in moisture availability and demand".

In addition, the inclusion of the figure above (correlation coefficients between ASCAT observables, radiation, precipitation and humidity) shows the correlation between the slope and curvature as a function of lag and how it varies among the ecoregions.

9. Lines 335-336:

"Strong temporal consistency was found between ASCAT backscatter and GRACE EWT, with the maximum backscatter coinciding with periods of maximum moisture availability." Please quantify this statement.

This statement summarizes the results in Figure 4, and the corresponding text. For clarity, the sentence has been reformulated as follows:

"The timing of the seasonal cycle of normalized backscatter was consistent with that of GRACE EWT, with the maximum (minimum) normalized backscatter coinciding with the maximum (minimum) EWT in all ecoregions."

10. Lines 336-337:

"Spatial patterns in mean and range of slope reflected spatial patterns in vegetation density." Please quantify this statement.

This has been revised to read "Spatial patterns in mean and range of slope reflect the ecoregions within the study area". This is based on the results presented in Figure 5, and the corresponding text in the results section. The following figure will be included in the supplementary materials to illustrate the difference in mean and range normalized backscatter, slope and curvature among the evergreen forest, flooded forest and savanna areas:

[Figure]

Figure A2 The mean and range of backscatter, slope and curvature for the ecoregions of interest.

11. Lines 350-351:

"Temporal consistency between the curvature and meteorological data suggests sensitivity to events such as litterfall and leaf flushing." How solid is this finding? Are their dates and periods reported

where litterfall or leaf flushing happened? Please try to show more content how the authors arrive at this finding.

This is discussed in lines 220-223 where we refer to the studies of Borchert et al. (2015) and Wagner et al. (2016). We have edited these lines, to emphasize the link between the seasonal cycle in curvature and those of radiation and humidity (i.e.) insolation and evaporative demand, and to mention that the peak in curvature occurs in July :

"In the Amazon rainforest, Borcher et al. (2015) observed that leaf flushing and flowering in adult trees of numerous species coincided with the rise and decline of insolation. Wagner et al. (2016) made a similar observation about leaf flushing and rising insolation in July, and also noted that the litterfall peak occurs when evaporative demand is highest and can persist through the dry season. Figure 6 (b) shows that although the changes in curvature are very small in the rainforest, the peak occurs in July on the rising limb of the radiation data, and when the specific humidity is near its minimum. Figures 6 (b-f) and Fig. A1 show the strong correspondence between curvature and radiation (positive correlation at a lag of 2 months) and specific humidity (negative correlation with a lag of -1 month) and that the highest values of curvature generally correspond to lower humidity, higher solar radiation and lower precipitation. This suggests that higher values of curvature may be related to litterfall during periods of high evaporative demand."

Lines 350-351 have also been revised as follows:

"The highest values of curvature coincide with periods of high evaporative demand (e.g. high radiation, lower humidity and lower precipitation). This suggests a link between curvature and phenological changes such as leaf flushing and litterfall. For example, the curvature peak in July in the rainforest occurs during rising insolation, and coincides with leaf flushing."

12.  Lines 355-357:

"Diurnal differences in backscatter during the dry season are dominated by transpiration losses. Long-term monitoring of these diurnal differences could provide insight into moisture availability and its influence on transpiration and vegetation functioning." Can this really be concluded with the presented analyses? Please add some reference or explanation.

We've added a reference to a recent publication (Konings et al. (2021) on forest dynamics that highlights the potential value of diurnal/diel differences in microwave observables in monitoring plant water status and drought response.

13.  Lines 366-368:

"… by vegetation structure and water content, and interactions between the soil and vegetation is essential to improve our ability to interpret and optimally use VOD derived from ASCAT." Is there forward modelling on VOD from ASCAT? Maybe even a sensitivity study? Please add references or a statement of future work at this point in the manuscript.

We added the following sentence to make it clearer that the study presented here was conducted as part of the continued development of ASCAT VOD products:

 "Therefore, this research contributes directly to the continued development of the ASCAT VOD products".

**Technical Comments:**

1. 15 & 16 should be placed within the section where they refer to and before starting of the next section (conclusions). Please review the document for further "late appearance" of figures.

Done. The final placement of figures will be determined in production, conforming to HESS norms.

2. Figure 16 caption: lines are dashed-dotted and not dotted. Please adapt caption text.

Done

**References:**

Birrer, I. J., E. M. Bracalente, G. J. Dome, J. Sweet, and G. Berthold. "σ signature of the Amazon rain forest obtained from the SeaSat scatterometer." *IEEE Transactions on geoscience and remote sensing* 1 (1982): 11-17.

Borchert, Rolf, Zoraida Calle, Alan H. Strahler, André Baertschi, Robert E. Magill, Jeremy S. Broadhead, John Kamau, Julius Njoroge, and Catherine Muthuri. "Insolation and photoperiodic control of tree development near the equator." *New Phytologist* 205, no. 1 (2015): 7-13

Frison, P-L., and Eric Mougin. "Use of ERS-1 wind scatterometer data over land surfaces." *IEEE Transactions on Geoscience and Remote Sensing* 34, no. 2 (1996): 550-560.

Hawkins, R., E. Attema, R. Crapolicchio, P. Lecomte, Josep Closa, P. J. Meadows, and S. K. Srivastava. "Stability of Amazon Backscatter at C-Band: Spaceborne Results from ERS-1/2 and RADARSAT-1." In *SAR workshop: CEOS Committee on Earth Observation Satellites*, vol. 450, p. 99. 2000.

Kennett, Rosemary G., and Fuk K. Li. "Seasat over-land scatterometer data. II. Selection of extended area and land-target sites for the calibration of spaceborne scatterometers." *IEEE Transactions on Geoscience and Remote Sensing* 27, no. 6 (1989): 779-788.

Khabbazan, S., S. C. Steele-Dunne, P. Vermunt, J. Judge, M. Vreugdenhil, and G. Gao. "The influence of surface canopy water on the relationship between L-band backscatter and biophysical variables in agricultural monitoring." *Remote Sensing of Environment* 268 (2022): 112789.

Konings, A. G., Saatchi, S. S., Frankenberg, C., Keller, M., Leshyk, V., Anderegg, W. R., ... & Zuidema, P. A. (2021). Detecting forest response to droughts with global observations of vegetation water content. *Global change biology*, *27*(23), 6005-6024.

Townsend, P. A. "Relationships between forest structure and the detection of flood inundation in forested wetlands using C-band SAR." *International Journal of Remote Sensing* 23, no. 3 (2002): 443-460.

Vermunt, Paul C., Susan C. Steele-Dunne, Saeed Khabbazan, Jasmeet Judge, and Nick C. van de Giesen. "Reconstructing Continuous Vegetation Water Content To Understand Sub-daily Backscatter Variations." *Hydrology and Earth System Sciences Discussions* (2021): 1-26.

Vermunt, Paul C., Saeed Khabbazan, Susan C. Steele-Dunne, Jasmeet Judge, Alejandro Monsivais-Huertero, Leila Guerriero, and Pang-Wei Liu. "Response of Subdaily L-Band Backscatter to Internal and Surface Canopy Water Dynamics." *IEEE Transactions on Geoscience and Remote Sensing* (2020).

Wagner, Fabien H., Bruno Hérault, Damien Bonal, Clément Stahl, Liana O. Anderson, Timothy R. Baker, Gabriel Sebastian Becker et al. "Climate seasonality limits leaf carbon assimilation and wood productivity in tropical forests." *Biogeosciences* 13, no. 8 (2016): 2537-2562.

---

## Author Response (AR1)

We thank the reviewers for their careful consideration of our manuscript and their constructive feedback. Our point-by-point response is provided below. The reviewer comments are shown in black. Our response is shown in blue. The line numbers refer to the **revised** manuscript.

A marked-up manuscript is also provided for your convenience. Note that there are minor discrepancies in line numbers between the marked-up and revised manuscript. However, it provides a convenient overview of the changes implemented.

**Reviewer Comments 1**

(https://hess.copernicus.org/preprints/hess-2021-406/#RC1)

This paper analyses the potential of radar data to monitor the seasonal cycle of vegetation and its water status for different biomes located in the Amazon basin using ASCAT C-band data. The paper is clear, well written and correctly organized. The results are interesting, physically sound in relation with the radar physics. My comments (see below) are really minor.

*Abstract*

Last line of the abstract, VOD is not mentioned before.

Thank you for this comment, we have adjusted this (See Line 21).

*Introduction*

L.24 much earlier reference exist on the sensitivity of microwave to the plant water content and status

Indeed, we have added some of the earliest reference to our knowledge: (Attema and Ulaby, 1978; Jackson et al., 1982; Owe et al., 2001). See Line 24. The reference to Konings et al., 2019 was made as this is an overview paper with the objective to: "provide an overview of the opportunities and pitfalls for using microwave observations for ecological studies".

L25-26: same remark

We have added the following references which we think cover a wider range of papers (Andela et al., 2013; Chaparro et al., 2019; Ferrazzoli et al., 1992; Liu et al., 2013; McNairn et al., 2000; Rao et al., 2019; Saatchi et al., 2013; Tian et al., 2016; Wagner et al., 1999). In addition, the Steele-Dunne et al., 2017 is a review paper on radar remote sensing for agricultural applications, containing many relevant references.  See Lines 25-27.

L.26: VOD is derived from passive microwave instrument

Although VOD is mostly retrieved from passive microwave instruments, VOD has also derived from active microwave observations, including Metop ASCAT (Liu et al., 2020; Vreugdenhil et al., 2016) and Sentinel-1 SAR (El Hajj et al., 2019).

 *ASCAT data processing*

Backscatter from the three beams are not acquired with the same azimuth angle while, azimuth effects can occurs depending on the canopy geometry. Could you elaborate on this ?

Backscatter is normalized for azimuthal effects according to (Bartalis et al., 2006). Here, biases as a result of azimuthal anisotropy are normalized by calculating a statistically based correction method based on historical backscatter observations over a period of three years. Azimuth effects are small over tropical forests, as was demonstrated by Bartalis et al., 2006 so we do not expect large effects of azimuth angle. The interested reader is referred to Steele-Dunne et al. (2019) for details of ASCAT processing in Line 130.

Figure 1: I don't see any mangrove on the LC map. Could perhaps be withdrawn from the legend ?

The map has been revised to make these more visible. We prefer to leave mangrove in the legend to acknowledge that they are present, even though they are very limited in extent.

*Results*

Figure 4 à please use the same range of value for the y-axis to compare more easily the seasonal dynamics and the amplitude of the seasonal signal / provide the legend for fig 4a

Figure 5 same remark as for figure 4

To compare the seasonal dynamics and amplitude we combined all regions in Figures 4a and 5a.

All ecoregions are shown on a single y-axes, with simplified symbology in Figures 4a and 5a to highlight the contrast between the amplitude and seasonal dynamics of the evergreen forest areas and the other ecoregions. However, in order to better analyze the seasonal signal in relation to environmental variables we then split them out per region with different y-axes. To make this more clear we will change the captions from (e.g.) : "Figure 4 Climatology of backscatter (green line), precipitation (bars), and EWT (blue line) for different cover types" to:

"Figure 4: Climatologies of backscatter for all ecoregions; five evergreen forest (dark green), flooded forest (cyan) and  three savanna (light green) (a). Plot (b) to (f) show climatology of backscatter (green line) with precipitation (bars) and EWT (blue line) per ecoregion. Note the different y-axes and that only the Jurua-Purus moist forest (fC) is shown as it is similar to the other evergreen forests."

"Figure 5: Climatologies of slope for all ecoregions; five evergreen forest (dark green), flooded forest (cyan) and  three savanna (light green) (a). Plot (b) to (f) show climatology of slope (green line) with precipitation (bars) and specific humidity (blue line) and radiation (red line) per ecoregion. Note the different y-axes and that only the Jurua-Purus moist forest (fC) is shown as it is similar to the other evergreen forests."

"Figure 6: Climatologies of curvature for all ecoregions; five evergreen forest (dark green), flooded forest (cyan) and  three savanna (light green) (a). Plot (b) to (f) show climatology of curvature (green line) with precipitation (bars) and specific humidity (blue line) and radiation (red line) per ecoregion. Note the different y-axes and that only the Jurua-Purus moist forest (fC) is shown as it is similar to the other evergreen forests."

Furthermore, we will add the following sentence to the text introducing Figure 4 (Lines 196-198) : "As the evergreen forest ecoregions showed very similar climatologies, only the Jurua-Purus moist forest is shown as a separate plot."

Fig 7 (right) à please provide a different color for the ocean (dark blue is used both for ocean and fraction at the ASCAT pixel scale). White such as in Figure 8 would be fine.

Thank you for this comment, we have changed this accordingly.

3.1.1. Cerrado analysis: The observed lower backscatter values occurring simultaneously with the peak of the slope (i.e. flatter backscatter response with regards to incidence angle) is not straightforward to me. From what I understand, photosynthetic activity is occurring after the wet season because of the radiation increase and because of the capacity of the plant to extract water in the deeper soil layers explaining why the volume diffusion is higher at this time (flatter backscatter response as a function of incidence angle). Dry season is also associated to dry upper soil conditions leading to lower backscatter level along the whole range of incidence explaining why the average backscatter levels are observed during the dry season. Am I right ? If yes, the section could be slightly rewritten to make it clearer.

Yes indeed, this is what is meant. We have rearranged the text and added the text indicated in red:

[revised manuscript text omitted]

3.3. Drought of 2010 and 2015: why didn't the authors had a look to the impact of drought on the diurnal differences of backscatter ? Would it be possible to provide as supplementary material for instance, the time series of the diurnal differences for both drought years ?

This analysis was performed. However, there was no significant spatial or temporal anomaly in the diurnal differences during the drought years. The results are provided below for your information. A comment to this effect has been included in Section 3.3 of the revised manuscript (Line 351-352): "No significant spatial or temporal anomalies were observed in the diurnal differences in backscatter during the drought years."

[Figure]

Figure 1: Diurnal difference in backscatter during the 2010 drought.

[Figure]

Figure 2: Diurnal difference in backscatter during the 2015 drought.


   We have done this. However, there were multiple issues we encountered and therefore decided not to put the maps in the manuscript. First, it is important to note that EWT is derived from GRACE, which has a spatial resolution on the order of hundreds of kilometers, and is based on measurements collected over a monthly period. Furthermore, EWT includes all contributions to terrestrial water storage including soil moisture, fresh biomass, but also groundwater. Spatial or temporal statistics between EWT and ASCAT observables would contain artefacts of this mismatch in temporal and spatial resolution, and sensing volume. Therefore, correlation statistics between ASCAT observables and EWT are of questionable value.

   Precipitation, radiation and humidity are at a higher spatial resolution and correlation statistics between them and the ASCAT observables can provide additional insights. However, as there are phase differences between the observables and the environmental variables a spatial correlation

map is not very informative. Therefore, we have performed a correlation analysis with different lag times between the observables and meteorological variables. The figure below illustrates the correlation coefficients averaged per region for different lag times between backscatter, slope and curvature with precipitation, radiation and humidity. This figure has been added in the supplementary material (Fig. A2) to the manuscript and will include the following text in Section 3.

For backscatter, Lines 203-206 now read: "Figure A2 shows temporal correlation between backscatter and precipitation is low for all ecoregions. A strong negative correlation and strong positive correlation are found with radiation and humidity for lags between -2 and 2 months, indicating that backscatter is lowest during drier periods with higher radiation and lower specific humidity. "

For slope, Lines 212-240 now read: "In Fig. 5(b-f), the seasonal cycle of slope in each ecoregion is compared to the corresponding cycles of radiation, specific humidity and precipitation which drive photosynthetic activity in the region. Note again that only the Jurua-Purus moist forest is shown as a separate plot. Furthermore, Fig. A2 illustrates the temporal correlation between slope and precipitation, radiation and specific humidity. In the Jurua-Purus moist forests (Fig. 5(b)), the change is slope is one-tenth that observed in the other ecoregions. The variations in radiation and specific humidity are also very limited. Nonetheless, the seasonal cycle of the slope follows that of the radiation with a lag of about 30 days (Fig. A2, R=0.75 at lag -1). This can be explained by the fact that the vegetation phenology in this tropical evergreen forest is driven by radiation (Romatschke and Houze Jr, 2013). The photosynthetic capacity depends on the available solar energy (Borchert et al., 2015). Energy availability drives transpiration and the accumulation of leafy biomass. This increases volume scattering from the canopy and therefore leads to an increase in the slope. Similar results were observed for the other forest ecoregions. In the Marajo varzea flooded forest (Fig. 5(c)), the variation in slope is much larger, and the seasonal cycle is clearly out of phase with that of the radiation. The seasonal variations in slope in this ecoregion are dominated by the influence of surface flooding rather than vegetation water content variations (Sect. 3.1.2).

In the Cerrado (Fig. 5(d)), there is a significant variation in specific humidity, and radiation as well as a strong seasonal cycle in precipitation. The peak in slope occurs during the driest time of year, when radiation is at a maximum and specific humidity and precipitation are at a minimum. Recall from Fig. 4, that this is also during the minimum EWT and backscatter period. This is also illustrated in Fig.A2 where strong negative correlations are found between slope and humidity. Correlations between slope and radiation are lower, and the highest correlation occurs at a lag of two months, i.e. slope leads radiation. Section 3.1.1 provides a detailed analysis of the vegetation types within the Cerrado ecoregion to better understand these variations. The slope values in the Guianan Savanna (Fig. 5(e)) are the lowest observed in all ecoregions, and also have the smallest variations among the non-forest cover types which are not strongly related to precipitation, radiation or specific humidity. This is consistent with the relatively low, but stable vegetation density associated with grasslands (Steele-Dunne et al., 2019). In the Beni Savanna (Fig. 5(f)), on the other hand, slope varies as much as in the Cerrado, and there is a very clear relationship between the slope and the atmospheric forcing data (Fig. 5 (f)). The maximum slope occurs at the peak of precipitation, EWT (from Fig. 4) and humidity. The minimum slope occurs during the dry season at the minimum in precipitation, humidity and EWT. This is consistent with the interpretation of slope as an indicator of vegetation density as the vegetation cover in this savanna changes dramatically in response to atmospheric forcing. This is also illustrated in Fig.A2, where high correlations are observed between slope and humidity with small lags. The contrast in the seasonal cycles in slope in Fig. 5 reflect the diversity of the vegetation cover types in the ecoregions and their varied response to moisture supply and demand."

For curvature, lines 251-255 now read:

"Figures 6 (b-f) and Fig. A2 show the strong correspondence between curvature and radiation (positive correlation at a lag of 2 months) and specific humidity (negative correlation with a lag of -1 month) and that the highest values of curvature generally correspond to lower humidity, higher solar radiation and lower precipitation. This suggests that higher values of curvature may be related to litterfall during periods of high evaporative demand."

[Figure]

Figure 1A: Temporal correlation between ASCAT observations (normalized backscatter (a-c), slope (d-f) and curvature (g-i)) and precipitation (left column), radiation (middle column) and humidity (right column).

3.      There is a lack of direct validation as backscatter, slope and curvature are "low level observables" concerning water dynamics in plants and cannot directly serve as vegetation parameters: Anyhow is a validation somehow, even in a future setup, possible? Please elaborate or discuss how a first-order validation could be conducted, potentially in an add-on study within a controlled environment. An improved understanding of the slope and curvature and how they are affected by environmental factors, here water dynamics in the Amazon, is needed to fully exploit the potential of the method. Curvature and slope are no direct indicators of plant density, phenology and structure. This is hard to link directly. Can we have an easier link? The reviewer likes to foster

more discussion and outlook kind of statements in the later sections of the manuscript showing how to overcome the limited understanding of the spatio-temporal dynamics of slope/curvature compared to the environmental ones.

We agree. A lot of our understanding of the incidence angle dependence of backscatter is based on experiments with tower-based or airborne radar systems conducted in the 80s and 90s to optimize the design of spaceborne radar systems. However, these experiments were generally focused on classification, soil moisture or biomass/LAI retrieval. Radar data were limited in space and/or time, and water dynamics (beyond soil moisture) were not considered. A recent study from Kim et al. (2015) provided some insight into factors influencing incidence angle dependence of L-band backscatter based on airborne radar data. In light of the increasing use of microwave data for vegetation applications, and increasingly related to water dynamics, we believe there is an urgent need for ground-based microwave experiments to improve our fundamental understanding of the links between microwave observables and water dynamics, and how microwave satellite remote sensing can optimally be used to observe vegetation water dynamics. This is the main goal of our field-based experimental research in which we are actively engaged (see Khabbazan et al. 2022, Vermunt et al. 2021, Vermunt et al. (2020) etc.). That said, one of the limitations of field-based experimental campaigns is that they are very localized. ASCAT provides more than 10 years of global, spaceborne radar data. In addition to ground-based experimental research, we think studies like the one presented here, to explore ASCAT dynamic vegetation parameters and explain the variations in terms of modeled or observed geophysical variables are equally valuable because they allow us to study a wide range of cover and climate types and the impact of events such as drought. Based on the results presented here, and the planned SCA instrument on Metop-SG, we would argue that incidence angle variations should be considered as a potentially valuable source of useful information. So, in any first-order ground validation, we would advocate the inclusion of incidence angle dependence. These arguments have been included in Lines 422 to 436 of the revised manuscript.

4. Another fundamental question is: How much are backscatter, slope and curvature correlated in space and time? How much can be simply explained by only backscatter? This may have been addressed before (maybe in Steele-Dunne et al., 2019), but a statement/paragraph would be beneficial to justify the analysis of the derivatives (slope, curvature). This could be also supported by EM modelling efforts.

The slope and curvature provide insight into the relative dominance of surface scattering versus volumetric and multiple scattering. This information is not contained in normalized backscatter alone. The relationship between backscatter, slope and curvature varies per cover type, as shown in the current study (compare and contrast the flooded forest to the cerrado and evergreen forest, for example). The following lines have been included in the introduction to make it clearer that the slope and curvature contain information that is not contained in normalized backscatter alone. The following sentences have been inserted in Lines 58-62 of the revised manuscript:

"In other words, slope and curvature are calculated and used to account for the influence of vegetation in the soil moisture retrieval. An increase in soil moisture results in an increase in backscatter at all incidence angles, while a change in the vegetation (due to growth cycle or water status) changes the sensitivity of backscatter to incidence angle, i.e. it results in a change in slope and curvature. So, the slope and curvature provide complementary information to the normalized backscatter."

5. Another point to discuss: How far is the presented water dynamics analysis transferable from Amazonas to somewhere else, e.g. other climates/biomes and regions of the world? More

explanation would help to shape the potentials and limitations of the approach in the discussion section.

We included a variety of ecoregions to illustrate that the relation between slope and curvature and the atmospheric forcing data varies according to land cover and climate. This also demonstrated the potential of using slope and curvature to analyze vegetation water dynamics in multiple cover types. This had been demonstrated by Steele-Dunne et al. (2019), who showed slope and curvature show useful information on vegetation water dynamics in a study limited to grasslands in the central US. Of course, we realize that the Amazonas is a very specific region, and that in other climates new challenges will arise. The following sentences have been added to the conclusion in lines 419-422:

"The current study was performed over different land cover types, demonstrating the potential to study vegetation water dynamics with these observables over different regions. However this research also confirms the need for further research to overcome the limited understanding of the spatio-temporal dynamics of slope compared to environmental drivers and effects in structure of vegetation."

6. Concerning vegetation penetration one major point is when the C-band EM waves start to interact with anything but not vegetation, like soil under vegetation: How far are soil influences on the backscatter signal playing a role, especially for lower vegetated or dry areas (e.g. Cerado)? Please evaluate and discuss potential non-vegetation influences on the signal, like soil scattering. Is there a criterion or threshold-based approach to find and exclude regions and/or times when non-vegetation effects, like from soil, have a too distinct/significant influence?

Microwave interactions with vegetated surfaces are complex due to the variety, in terms of size and dielectric properties, of the vegetation constituents and the influence this has on the propagation of microwaves through the vegetation, interactions within the vegetation and interactions between the soil and vegetation. We always consider the total backscatter as a combination of contributions from a soil-vegetation continuum. The rationale behind using the slope, for example, is that an increase in slope is indicative of a transition in behaviour between predominantly surface scattering (from the soil) to predominantly volume or multiple scattering (from the vegetation) and vice versa. This tells us if the normalized backscatter variations are due to soil only, vegetation only, or some combination of both. Naturally, for areas or periods when slope is low, there may be a contribution of soil scattering to the total backscatter signal. This is also reflected in the correspondence between EWT and backscatter, since EWT is an aggregated signal including soil and vegetation water among other elements. Thus, we agree that in terms of backscatter dynamics there might be added-value in further analyzing this, however the main aim of this research was to investigate the backscatter incidence angle relationship.

Working with the slope values allows us to observe the physical process of fresh biomass change (either due to phenology or water status) through its effect on the backscatter incidence angle relationship. As this is, in itself, an indicator of which type of scattering contributes to the signal, and these are continuous processes, we do not see the added-value of introducing criteria or thresholds to categorize behaviour rather than use the information directly.

6.      There is a spatio-temporal scale gap as well as a sensing volume gap (C-band EM wave penetration vs. 3D gravity field dynamics) between GRACE EWT and ASCAT observations. Hence, the reviewer has doubts that (lines 181-183)"…in each ecoregion, there is clear agreement between the seasonality of EWT and backscatter. This indicates that backscatter is influenced by moisture availability in terms of total terrestrial water storage, which includes groundwater storage." This is a

strong statement and "a clear agreement" is not really statistically quantified. Please add some statistical or more quantitative analysis for justification of this agreement. Moreover, please explain and/or discuss the scale gap and sensing volume gap of the two remote sensing observations.

There is an obvious scale and sensing volume mismatch between the two datasets. This is why we do not consider it sound to provide a statistical or quantitative comparison between the two. It would be dominated by artefacts of the difference in spatial and temporal scale between the two products. We have included the following text in Section 2.3 by way of explanation.

Lines 156-158: "Precipitation, radiation and humidity are hypothesized to be the main atmospheric forcing for vegetation activity in the Amazon (\citep{nemani_climate-driven_2003}). Therefore, these three forcings are compared to slope and curvature. As they are on similar temporal and spatial scale quantitative comparisons are performed."
Lines 163-165: " EWT includes variations in all terrestrial water storage terms including groundwater, soil moisture, vegetation, and surface water. Therefore, EWT is only qualitatively compared to backscatter, which is affected by soil moisture and vegetation. "

Minor Comments:

1. Lines 171-172:

"The Guianan savanna, with sparse vegetation, has low mean slope values. The Cerrado, on the other hand, shows mean values higher than the evergreen forests. This is unexpected since slope is generally considered a measure of "vegetation density", and the evergreen forests are much denser than savannas."

The forest/vegetation density that microwaves "see" can be twofold. Density can come from dry biomass/structure, which is dry matter based, or come from vegetation water, which is wet matter based. This comment may help to review the above-mentioned paragraph.

Based on this comment, we need to make it clearer that when we discuss "vegetation density", we are not referring to plants per area, but actually to above ground fresh biomass which is indeed a combination of dry biomass and vegetation water content. This has been made clearer by including the following statement in lines 66-67: "Slope is considered an indication of vegetation density, or above ground fresh biomass, which is a combination of dry biomass and vegetation water content."

2. Figures 5 & 6:

These Figures contain four y-axes and show an overview how the incorporated parameters/variables behave along time. This is a first overview along time. What is missing is a statistical evaluation of the temporal correlation.

In response to this comment, the following figure has been added to the supplementary materials. It shows the correlation between normalized backscatter, slope and curvature and the precipitation, radiation and humidity as a function of lag. This makes it clear that humidity and radiation are more important drivers than precipitation in this region. We have also included text on this, which is

described in detail in response to major comment 2.

[Figure]

3.      Lines 258-263 and lines 351-352:

"This is due to multiple scattering between the water surface and the vegetation." In terms of scattering mechanism characterization: Should this be double bounce scattering (water-vegetation)? What kind of scattering mechanism could this be?

Under forest/woody vegetation, this is a combination of double bounce scattering between the surface and trunks, and multi-path scattering between the surface and the vegetation (Townsend (2002). This has been clarified in the revised text and the reference has been added. See lines 297-298.

4.      Lines 261-263:

"…the curvature changes considerably and even changes sign during the flooded period. This illustrates that the curvature includes useful information on changes in the scattering mechanisms, which are related to physical changes at the land surface." Is there a way to link the curvature more directly to the physical variables? Could the authors try modeling or anything similar? It would be interesting to couple a forward model with the slope and curvature metric to investigate sensitivities and dependencies. Could references (if done) or an outlook statement (if not yet done) be included in the manuscript?

In other research, we have developed a data-driven approach to simulate the ASCAT observables based on land surface variables (Xu Shan et al., RSE (in review)). Alternatively, Radiative Transfer Modelling could be used to simulate C-band backscatter as a function of incidence angle to quantify the sensitivity of slope and curvature to physical changes at the land surface. We have added the following statement in lines 434-436:

"Ongoing research is focused on using data-driven and radiative transfer modeling approaches to investigate the sensitivity of slope and curvature to physical changes at the land surface."

5. Figures 12 & 13:

Can Figures 12 and 13 be shown in a way that they are jointly together and their similarity or difference in pattern can be understood intuitively? Maybe an add-on figure might be an option or a replacement of Figs. 12 & 13. For most of the domain, especially the evergreen forests, high values in EWT coincide with negative diurnal differences in backscatter and vice versa. In the moment, the comparison of two 6-pannel figures (12 & 13) appears complicated.

We have combined the two Figures into a single one to facilitate visual comparison of EWT and diurnal difference in backscatter. See Figure 12 of the revised manuscript.

6. Figure 14:

Figure 14(a) is indicative of the seasonal variations observed across the evergreen forest ecoregions. Note that the diurnal differences are very small (< 0.06 dB). These seem to be really small differences.

How about signal stability in terms of radiometric resolution? i.e. How noise-prone are these subtle differences? Please add an explanatory paragraph and some discussion about this point.

Refer to our response to major comment 2. The variability in backscatter from evergreen forest ecoregions is extremely limited. In fact, the Amazon rainforest has long been used as a calibration target for spaceborne radar systems (Birrer et al. (1982); Kennet and Li (1989); Frison and Mougin (1996); Hawkins et al. (2000)). This has been included in lines 192 to 195 of the revised manuscript in the discussion of Figure 4.

7. Lines 294-296:

"One possible explanation for this unusual seasonal cycle *could be* that it is related to a change in the relative dominance of the forests and grasslands in the backscatter signal. The transition from positive to negative curvature values during the EWT peak *could* indicate an increased contribution from tree patches and shrubs during the wetter period." How can this be justified? This sounds quite speculative. Should the statement be softened?

It would be inappropriate to draw a stronger conclusion without a detailed modeling study, for which detailed ground data (structure, vegetation water content) would be needed. Hence, we were careful to use "could" to indicate that this is a possible explanation but not one that we are currently able to confirm.

8. Lines 331-332:

"However, the current study is the first to relate the spatial and temporal variations in slope and curvature to moisture availability and demand." Are we really seeing a relationship that is statistically significant? Please add more statistical (correlation) analyses to support the statement.

This sentence has been revised to read: "However, the current study is the first to attempt to explain the spatial and temporal variations in slope and curvature in terms of seasonal variations in moisture availability and demand".

In addition, the inclusion of the figure above (correlation coefficients between ASCAT observables, radiation, precipitation and humidity) shows the correlation between the slope and curvature as a function of lag and how it varies among the ecoregions.

9. Lines 335-336:

"Strong temporal consistency was found between ASCAT backscatter and GRACE EWT, with the maximum backscatter coinciding with periods of maximum moisture availability." Please quantify this statement.

This statement summarizes the results in Figure 4, and the corresponding text. For clarity, the sentence has been reformulated as follows:

"The timing of the seasonal cycle of normalized backscatter was consistent with that of GRACE EWT, with the maximum (minimum) normalized backscatter coinciding with the maximum (minimum) EWT in all ecoregions."

10. Lines 336-337:

"Spatial patterns in mean and range of slope reflected spatial patterns in vegetation density." Please quantify this statement.

This has been revised to read "Spatial patterns in mean and range of slope reflect the ecoregions within the study area". This is based on the results presented in Figure 5, and the corresponding text in the results section. The following figure is included in the supplementary materials (Figure A1) to illustrate the difference in mean and range normalized backscatter, slope and curvature among the evergreen forest, flooded forest and savanna areas:

[Figure]

11. Lines 350-351:

"Temporal consistency between the curvature and meteorological data suggests sensitivity to events such as litterfall and leaf flushing." How solid is this finding? Are their dates and periods reported where litterfall or leaf flushing happened? Please try to show more content how the authors arrive at this finding.

This was discussed in lines 220-223 of the original manuscript where we refer to the studies of Borchert et al. (2015) and Wagner et al. (2016). We have edited these lines (Lines 247-255 of the revised manuscript), to emphasize the link between the seasonal cycle in curvature and those of radiation and humidity (i.e.) insolation and evaporative demand, and to mention that the peak in curvature occurs in July :

"In the Amazon rainforest, Borcher et al. (2015) observed that leaf flushing and flowering in adult trees of numerous species coincided with the rise and decline of insolation. Wagner et al. (2016) made a similar observation about leaf flushing and rising insolation in July, and also noted that the litterfall peak occurs when evaporative demand is highest and can persist through the dry season. Figure 6 (b) shows that although the changes in curvature are very small in the rainforest, the peak occurs in July on the rising limb of the radiation data, and when the specific humidity is near its minimum. Figures 6 (b-f) and Fig. A1 show the strong correspondence between curvature and radiation (positive correlation at a lag of 2 months) and specific humidity (negative correlation with a lag of -1 month) and that the highest values of curvature generally correspond to lower humidity, higher solar radiation and lower precipitation. This suggests that higher values of curvature may be related to litterfall during periods of high evaporative demand."

In addition, Lines 394-397 of the revised manuscript now read:

"The highest values of curvature coincide with periods of high evaporative demand (e.g. high radiation, lower humidity and lower precipitation). This suggests a link between curvature and phenological changes such as leaf flushing and litterfall. For example, the curvature peak in July in the rainforest occurs during rising insolation, and coincides with leaf flushing."

   12.   Lines 355-357:

"Diurnal differences in backscatter during the dry season are dominated by transpiration losses. Long-term monitoring of these diurnal differences could provide insight into moisture availability and its influence on transpiration and vegetation functioning." Can this really be concluded with the presented analyses? Please add some reference or explanation.

We've added a reference to a recent publication (Konings et al. (2021) on forest dynamics that highlights the potential value of diurnal/diel differences in microwave observables in monitoring plant water status and drought response. See Line 403 of the revised manuscript.

   13.   Lines 366-368:

"… by vegetation structure and water content, and interactions between the soil and vegetation is essential to improve our ability to interpret and optimally use VOD derived from ASCAT." Is there forward modelling on VOD from ASCAT? Maybe even a sensitivity study? Please add references or a statement of future work at this point in the manuscript.

We added the following sentence (Lines 414-415) to make it clearer that the study presented here was conducted as part of the continued development of ASCAT VOD products:

 "Therefore, this research contributes directly to the continued development of the ASCAT VOD products".

**Technical Comments:**

   1.   15 & 16 should be placed within the section where they refer to and before starting of the next section (conclusions). Please review the document for further "late appearance" of figures.

Done. The final placement of figures will be determined in production, conforming to HESS norms.

2. Figure 16 caption: lines are dashed-dotted and not dotted. Please adapt caption text.

Done

Citation: https://doi.org/10.5194/hess-2021-406-RC2

---

## Author Response (AR2)

Dear Dr. Green,
Please find below our response to the reviewer comments. The comments of the reviewer are provided in Black. Our response is highlighted in Blue.

In addition, please note that we have moved Figures A1 and A2 from the Appendix to Supplementary Materials so that we have one single additional file containing all ancillary tables and figures including those requested by the reviewers.

**General Comment:**

The submitted manuscript presents the analysis of ASCAT time series data (backscatter, slope & curvature) over the greater Amazon region with regards to water dynamics and two drought events. Additional meteorological (e.g. precipitation from GPCP) and water dynamics (from EWT – GRACE) information are incorporated into the analyses for comparison. The following comments & suggestions are the remaining issues that have not been fully addressed in the last review round; Non-appearing comments from last review are considered as solved by the reviewer:

**Major Comments:**

1. The study analyses are based on very small changes in backscatter (sometimes well below 0.1 dB in variation). This puts a massive demand on radiometric stability (and NESZ) of the ASCAT sensor. Please elaborate on this topic and include justifying statements. How far are these small backscatter variations showing significant and stable correlations to variations in environmental properties in the Amazonian vegetation? Is there a lower limit in sensitivity? The reviewer thinks it would be reasonable to define a lower limit.

Answer authors:
Wilson et al. (2010) mention that ASCAT was expected to have an accuracy of +/-0.3dB at 95% confidence level. A subsequent validation study by Anderson et al. (2011) showed a calibration accuracy of 0.15 - 0.25 dB. Therefore, changes on the order of 0.1dB are unquestionably close to the limits of the sensor. It is important to note, however, that the radiometric accuracy is expected to be better (i.e. less noisy) over stable, homogeneous targets (e.g. evergreen rainforest). Furthermore, the results presented here have been averaged in space or time, or both, which also reduces the noise. Consequently, it is reasonable to assume that the spatial and temporal patterns observed can be attributed to geophysical variability rather than observation error. This point has been summarized in Lines 145-147.

Answer reviewer:
This point is still considered critical by the reviewer. The changes in the text are well received. However, the sentence in lines 145-146 "…reduce noise, the backscatter data is averaged in space (over the ecoregions of interest) and/or time (to monthly or decadal intervals)." is still a qualitative statement. Please quantify the averaging in space and time. More precise, please provide a table showing the number of equivalent looks that are used for averaging. It is required to understand how many samples were used to stabilize the backscattered signal.

Author response:
Tables S1 and S2 have been added in Supplementary Materials. The following has been included in the description of the ASCAT data: "To reduce noise, the backscatter data is averaged in space (over the ecoregions of interest) and/or time (to monthly or dekadal intervals). The number of grid points averaged is provided in Tables S1 and S2. Data are available every 1-2 days (Wagner et al. 2013)."

6. Concerning vegetation penetration one major point is when the C-band EM waves start to interact with anything but not vegetation, like soil under vegetation: How far are soil influences on the backscatter signal playing a role, especially for lower vegetated or dry areas (e.g. Cerado)? Please evaluate and discuss potential non-vegetation influences on the signal, like soil scattering. Is there a criterion or threshold-based approach to find and exclude regions and/or times when non-vegetation effects, like from soil, have a too distinct/significant influence?

Answer authors:
Microwave interactions with vegetated surfaces are complex due to the variety, in terms of size and dielectric properties, of the vegetation constituents and the influence this has on the propagation of microwaves through the vegetation, interactions within the vegetation and interactions between the soil and vegetation. We always consider the total backscatter as a combination of contributions from a soil-vegetation continuum. The rationale behind using the slope, for example, is that an increase in slope is indicative of a transition in behaviour between predominantly surface scattering (from the soil) to predominantly volume or multiple scattering (from the vegetation) and vice versa. This tells us if the normalized backscatter variations are due to soil only, vegetation only, or some combination of both. Naturally, for areas or periods when slope is low, there may be a contribution of soil scattering to the total backscatter signal. This is also reflected in the correspondence between EWT and backscatter, since EWT is an aggregated signal including soil and vegetation water among other elements. Thus, we agree that in terms of backscatter dynamics there might be added-value in further analyzing this, however the main aim of this research was to investigate the backscatter incidence angle relationship. Working with the slope values allows us to observe the physical process of fresh biomass change (either due to phenology or water status) through its effect on the backscatter incidence angle relationship. As this is, in itself, an indicator of which type of scattering contributes to the signal, and these are continuous processes, we do not see the added-value of introducing criteria or thresholds to categorize behaviour rather than use the information directly.

Answer reviewer:
As far as the reviewer understands from the answer of the authors, "...when the slope is low, there may be a contribution of soil scattering to the total backscattering signal...". Another statement says: "The rationale behind using the slope, for example, is that an increase in slope is indicative of a transition in behaviour between predominantly surface scattering (from the soil) to predominantly volume or multiple scattering (from the vegetation) and vice versa.". These two statements motivate that for regions with non-closed-canopy conditions and significant soil contribution, the water sensitivity of the slope

and curvature may be due to soil moisture dynamics rather than vegetation water ones. The reviewer agrees that this misfit in water dynamic sensitivity might be weakened, as the different water storage compartments (soil, vegetation) are linked by the soil-plant-atmosphere system. In the end, the reviewer would ask for including a paragraph in the discussion section to address this challenge of soil scattering contributions in the backscattered signal and how to deal with it.

Author response:
The following sentences have been included in the Conclusion section: "For regions with non-closed-canopy conditions and significant soil contribution, the water sensitivity of the slope and curvature may be influenced, or even dominated by soil moisture dynamics (Greimeister-Pfeil et al. (2022)). Furthermore, the various storage compartments (soil, vegetation) are linked by the soil-plant-atmosphere system."

"An improved physical understanding of the influence of both soil and vegetation on slope and curvature is essential. Future research should also include forward electro-magnetic modelling of multi-angular backscatter (i.e slope and curvature) to improve our understanding of how they relate to vegetation water and biomass variations as well as soil moisture."

6. There is a spatio-temporal scale gap as well as a sensing volume gap (C-band EM wave penetration vs. 3D gravity field dynamics) between GRACE EWT and ASCAT observations. Hence, the reviewer has doubts that (lines 181-183)"…in each ecoregion, there is clear agreement between the seasonality of EWT and backscatter. This indicates that backscatter is influenced by moisture availability in terms of total terrestrial water storage, which includes groundwater storage." This is a strong statement and "a clear agreement" is not really statistically quantified. Please add some statistical or more quantitative analysis for justification of this agreement. Moreover, please explain and/or discuss the scale gap and sensing volume gap of the two remote sensing observations.

Answer authors:
There is an obvious scale and sensing volume mismatch between the two datasets. This is why we do not consider it sound to provide a statistical or quantitative comparison between the two. It would be dominated by artefacts of the difference in spatial and temporal scale between the two products. We have included the following text in Section 2.3 by way of explanation. Lines 156-158: "Precipitation, radiation and humidity are hypothesized to be the main atmospheric forcing for vegetation activity in the Amazon (\citep{nemani_climate-driven_2003}). Therefore, these three forcings are compared to slope and curvature. As they are on similar temporal and spatial scale quantitative comparisons are performed."
Lines 163-165: " EWT includes variations in all terrestrial water storage terms including groundwater, soil moisture, vegetation, and surface water. Therefore, EWT is only qualitatively compared to backscatter, which is affected by soil moisture and vegetation. "

Answer reviewer:
Many thanks for all explanations and adaptions concerning sensing volume mismatch. Please also include the spatio-temporal scale gap/mismatch of GRACE-based EWT with

ASCAT-based backscatter & derivatives as a statement close to the statement in lines 163-165.

Author response:
Lines 165-170 have been revised and now read:
"Note that EWT includes variations in all terrestrial water storage terms including groundwater and surface water, in addition to the variables of interest in this paper, namely soil moisture and vegetation. Furthermore, EWT is based on monthly data with a spatial resolution of hundreds of kilometers. Statistical comparisons between the EWT and ASCAT would be strongly influenced by the sensitivity of EWT to ground- and surface water and by artefacts of the difference in spatial and temporal scale between the two products. Therefore, EWT is only qualitatively compared to backscatter, which is affected by soil moisture and vegetation."

**Minor Comments:**

6. Figure 14:
Figure 14(a) is indicative of the seasonal variations observed across the evergreen forest ecoregions. Note that the diurnal differences are very small (< 0.06 dB). These seem to be really small differences.
How about signal stability in terms of radiometric resolution? i.e. How noise-prone are these subtle differences? Please add an explanatory paragraph and some discussion about this point.

Answer authors:
Refer to our response to major comment 2. The variability in backscatter from evergreen forest ecoregions is extremely limited. In fact, the Amazon rainforest has long been used as a calibration target for spaceborne radar systems (Birrer et al. (1982); Kennet and Li (1989); Frison and Mougin (1996); Hawkins et al. (2000)). This has been included in lines 192 to 195 of the revised manuscript in the discussion of Figure 4.

Answer reviewer:
Please explain also in the text paragraphs referring to Figure 14 and also Figure 15 that these very small fluctuations in backscatter may only be scientifically evaluable in rainforest regions, where the spatio-temporal backscatter dynamics (radiometric variations) are the most stable in the world.

Author response:
The following statement has been added at the end of the discussion of Figure 15: "Note that the very small fluctuations in backscatter observed in Figures 14 and 15 may only be scientifically evaluated in rainforest regions, where the spatio-temporal backscatter dynamics (radiometric variations) are among the most stable in the world."

13. Lines 366-368:
"… by vegetation structure and water content, and interactions between the soil and vegetation is essential to improve our ability to interpret and optimally use VOD derived

from ASCAT." Is there forward modelling on VOD from ASCAT? Maybe even a sensitivity study? Please add references or a statement of future work at this point in the manuscript.

Answer authors:
We added the following sentence (Lines 414-415) to make it clearer that the study presented here was conducted as part of the continued development of ASCAT VOD products: "Therefore, this research contributes directly to the continued development of the ASCAT VOD products".

Answer reviewer:
Thank you for adding the sentence. However, it would be very interesting to understand if there are publications or even internal sensitivity studies on modelling or even estimating VOD from ASCAT. If these references exist, please add them to the paragraph.
Moreover, it would be of great interest (of the reviewer) if there exists motivation of the authors for a forward electro-magnetic modelling study of multi-angular backscatter derivatives (meaning slope and curvature) and how they are linked/influenced by vegetation dynamics (water & biomass). Or is this already existing somewhere? It could lead to a direct retrieval of variables of the vegetation dynamics (e.g. VWC or wet biomass or dry biomass) from slope and curvature.

Author response:
Refer to comment above regarding disentangling soil and vegetation effects and the recommendation to perform forward electro-magnetic modeling to improve understanding. In addition, the paragraph has been edited as follows:"
Slope and curvature may be influenced by the number and distribution of the scatterers, and their dielectric properties, all of which influence the optical depth i.e. the attenuation of the signal by the vegetation.Our improved understanding of the slope and curvature and how they are affected by vegetation structure and water content, and interactions between the soil and vegetation is essential to improve our ability to interpret and optimally use VOD derived from ASCAT. Therefore, this research contributes directly to the continued development of the ASCAT VOD products. For example, it provides further insights in the VOD calculated from ASCAT by Vreugdenhil et al. (2016), where the main temporal dynamics stem from the slope and curvature."